# Mathematical and Machine Learning Models for Groundwater Level Changes: A Systematic Review and Bibliographic Analysis

Stephen Afrifa [1,2,*], Tao Zhang [1], Peter Appiahene [2] and Vijayakumar Varadarajan [3]

1 Department of Information and Communication Engineering, Tianjin University, Tianjin 300072, China
2 Department of Computer Science and Informatics, University of Energy and Natural Resources, Sunyani 00233, Ghana
3 School of Computer Science and Engineering, University of New South Wales, Sydney, NSW 2052, Australia
* Correspondence: afrifastephen@tju.edu.cn; Tel.: +233-24-7498261

**Abstract:** With the effects of climate change such as increasing heat, higher rainfall, and more recurrent extreme weather events including storms and floods, a unique approach to studying the effects of climatic elements on groundwater level variations is required. These unique approaches will help people make better decisions. Researchers and stakeholders can attain these goals if they become familiar with current machine learning and mathematical model approaches to predicting groundwater level changes. However, descriptions of machine learning and mathematical model approaches for forecasting groundwater level changes are lacking. This study picked 117 papers from the Scopus scholarly database to address this knowledge gap. In a systematic review, the publications were examined using quantitative and qualitative approaches, and the Preferred Reporting Items for Systematic Reviews and Meta-Analyses (PRISMA) was chosen as the reporting format. Machine learning and mathematical model techniques have made significant contributions to predicting groundwater level changes, according to the study. However, the domain is skewed because machine learning has been more popular in recent years, with random forest (RF) methods dominating, followed by the methods of support vector machine (SVM) and artificial neural network (ANN). Machine learning ensembles have also been found to help with aspects of computational complexity, such as performance and training times. Furthermore, compared to mathematical model techniques, machine learning approaches achieve higher accuracies, according to our research. As a result, it is advised that academics employ new machine learning techniques while also considering mathematical model approaches to predicting groundwater level changes.

**Keywords:** machine learning; mathematical model; statistical model; climate; systematic review; groundwater; groundwater level



## 1. Introduction

Groundwater is one of the most important sources of water [1]. Groundwater provides drinking water to up to 50% of the world's population [2], as well as accounting for 43% of all agricultural water [3]. Groundwater is a major natural resource for our mother earth, accounting for around 95 percent of all freshwater on the planet [4], making it essential for human existence and economic advancement. The effects of global climate change on groundwater level changes are significant enough to warrant further investigation in order to enhance forecasting and future consequences [5]. Due to poor extraction and overexploitation, the volume of groundwater has declined in recent years, worsening future water shortages [6]. The need to study these important resources is vital to ensure sustainable development. Researchers have used a variety of machine learning (ML) models to predict groundwater level (GWL) changes, including [7], who used a hybrid ML model, ref. [8] who used an ensemble modeling framework based on spectral analysis, machine learning, and uncertainty analysis, ref. [9] who used two ANN models, ref. [10] who used

random forest (RF), and [11], who used two commonly employed machine learning models: multi-linear regression (MLR) and random forest (RF).

In addition, statistical models (SM) by [12–14] and mathematical models (MM) by [15–17] have also been used to predict GWL changes. In recent years, there has been a growing interest in applying machine learning and data-driven methodologies to groundwater modeling [18–20]. With the persistent threat of climate change and human influences, access to high-resolution and continuous hydrologic data is critical for projecting trends and water resource availability [19]. Many studies on groundwater level measurement have been conducted due to the relevance of assessing and forecasting groundwater level changes [20]. Predicting the level of groundwater in farms with accuracy and simplicity is an important part of agricultural water management [21,22]. The use of groundwater for various activities has increased in the last decades [8]. Machine learning and mathematical modeling methodologies and techniques [23] have been used to forecast changes in groundwater levels based on research [24,25]. In addition, it must be noted that the state of the water as well as its level are interesting aspects of groundwater modeling (e.g., concentrations of different chemicals and geospatial analysis) [26,27]. However, the combination of mathematical and machine learning approaches to forecast groundwater level changes is scarce. Although previous studies have summarized and addressed either ML or MM on GWL, none have, to our knowledge, focused on both ML and MM models on groundwater level fluctuations in a systematic means. The goal of this study is to offer a thorough evaluation of existing research on the application of machine learning, as well as mathematical and statistical methodologies, for modeling and forecasting groundwater resources. By conducting a detailed analysis of the gathered data in a systematic review, this work explores the applicability of ML, MM, and SM techniques for estimating groundwater level resources. Monitoring groundwater levels helps to ensure that the aquifer systems in the basin are well understood, as well as how they react to changes in groundwater recharge, pumping, and other factors [26]. The estimation of groundwater level variations can also be used to determine how much groundwater storage has increased as a result of recharge or decreased as a result of discharge, such as extraction for usage [27]. For effective groundwater resource development, quantification of groundwater level fluctuations is a fundamental prerequisite, and this is especially important for areas with a preponderance of semi-arid and arid environments. Understanding how the hydrogeologic system responds and understanding the effects of climate to conserve aquifers for irrigation, among other things, depend greatly on the findings of measuring groundwater levels.

"A systematic review tries to bring evidence together to answer a pre-defined research topic", say Pollock and Berge [28]. Despite the fact that the scope of this systematic review is broad, it concentrates on groundwater level change prediction and the performance of ML and MM techniques in terms of groundwater level attributes.

There is no doubt that ML and MM practitioners and researchers generally believe that AI has advanced [29], but to the best of our knowledge, these statements are hypothetical and have not been empirically supported. The majority of studies now in existence have either shown how their innovation outperforms a number of currently used techniques or have surveyed a sample of systems and evaluated their performance [30] in relation to others. The amount of selection biases is significant across the board. Additionally, this research represents the most recent work in the fields of ML and MM for simulating changes in groundwater levels. As a result, there is a need for a compiled literature that offers descriptions of the problems, difficulties, and potential future research areas.

The next section gives an overview of how ML and MM/SM approaches are used to anticipate groundwater level changes. This is followed by a discussion of previously published similar efforts as well as the current knowledge gap. The research technique is described, as well as the findings, comments, and conclusions.

## 2. Background Literature

The literature on machine learning and mathematical model techniques for simulating groundwater level changes is presented in this area. The section also includes research that shows how both ML and MM approaches were used to predict GWL fluctuations, as well as literature that exclusively shows modular finite-difference groundwater flow (MODFLOW) approaches. In addition, relevant works (reviews) on our subject are discussed.

### 2.1. Machine Learning for Groundwater Level Change Prediction

The resurgence of attention in machine learning is due to the growing volumes and types of available data, less expensive computational handling, and all-the-more ground-breaking and modest data storage [31]. In recent years, academics have created a number of machine learning algorithms for forecasting stock trends [32] and groundwater level prediction [33], for example. Today, even on a massive scale, it is possible to quickly and organically develop models that can dissect larger, more mind-boggling data and deliver faster, more exact results. The ability of machine learning, which has the capability to learn complex correlations between the important hydrogeological factors (HGFs) and the groundwater level (GWL), is essential [34]. There are several machine learning algorithms available, including classification and regression models. Artificial intelligence (AI) models have been extensively used in the last 20 years to overcome the limitations of traditional numerical models for GWL simulation [24]. Therefore, improving the planning and management of water resources requires the use of precise soft computing methods for groundwater level (GWL) predictions.

Forecasting groundwater levels (GWL) is critical for irrigation planning, water supply, and land development. Although some studies have used artificial neural networks (ANN) [35], support vector machine (SVM) [36] and random forest (RF) [37] methods, support vector machine, generalized regression neural network, convolutional neural network, long short-term memory, and gated recurrent network are examples of machine learning and deep learning methodologies used by others [38] to simulate GWL changes. For example, ref. [11] employed multi-linear regression (MLR) and random forest (RF) methods. ANN models [35] with one natural factor and two anthropogenic factors as input variables were used to anticipate groundwater levels. Furthermore, ANN, SVM, and extreme gradient boosting (XGB), three frequently used machine learning models, were tested for their efficacy in predicting groundwater levels [39].

Ensemble modeling is also a great method to improve the presentation of models. Ensemble machine learning is used to predict GWL changes [40]; however, it is not always the most portable of the ML models [2]. Using a variety of modeling algorithms or training data sets, ensemble modeling is the process of building numerous varied models to predict an outcome [2]. By mixing numerous models instead of just one, ensemble methods offer methodologies that try to increase the accuracy of outcomes in models [36]. The combined models considerably improve the results' accuracy. Due to this, ensemble approaches in machine learning have gained prominence. The goal of employing ensemble models is to lower the prediction's generalization error. When using the ensemble approach, the prediction error lowers as long as the basis models are diverse and independent. Artificial intelligence methods [41], such as genetic programming (GP) and adaptive neural fuzzy inference system (ANFIS) [42], and deep learning models [43], such as LSTM and developed LSTM extension (DeepAR), help in groundwater level forecast and simulation. Furthermore, ref. [44] developed an ensemble of one- to five-month lead-time estimates for water tables based on various data-driven models (DDMs). Furthermore, ref. [6] used a hybrid ANN model to forecast GWL. Models that explicitly mix two or more models are known as hybrid models [45]. The ML architecture that underlies hybrid machine learning (HML) algorithms differs slightly from the conventional workflow. It is understood that every ML algorithm has a strategy for determining the optimal model in the context of an ideal configuration.

The radial basis function (RBF) neural network–whale algorithm (WA) model, the multilayer perception (MLP) model, and genetic programming (GP) were the three ANN models used in the same way [46,47] used ANN and ANFIS for prediction of GWL. In another instance, refs [2,48] proposed a new machine learning ensemble model (ARZ ensemble); thus, automatic multilayer perceptron (AutoMLP), RF, and ZeroR based on a majority voting-based technique applied to its standalone classifier. Because of its capacity to simulate nonlinearities between GWL and its drivers (e.g., rainfall), machine learning (ML) (e.g., artificial neural networks) is increasingly being used to anticipate GWL [49–51]. Machine learning has been used to supplement existing technical methods because it provides effective standards, increases efficiency, and improves GWL prediction performance [50].

Despite all of ML's advantages, the continuous growth of techniques makes it difficult for academics to determine the most effective technique and its impact on GWL prediction. Machine learning has already been used to predict GWL in a number of studies, but there is a need for a collected literature that outlines the domain's issues, challenges, and future research aims.

### 2.2. Mathematical/Statistical Modeling for Groundwater Level Change Prediction

The process of turning issues from an application zone into manageable mathematical formulations utilizing a hypothetical and arithmetical analysis to provide perception, answers, and guidance for application development is known as mathematical modeling [51]. Numerical models are powerful tools for simulating and analyzing groundwater dynamics under varying conditions, and they are employed all around the world [16]. In the literature, mathematical models (MM) have also been used to predict groundwater level variations. From 1993–2019, everyday groundwater level readings were analyzed statistically by [52]. In a work by [4], the stability theory of nonlinear differential equations was applied to create a mathematical model to optimize the groundwater level declination using a set of nonlinear ordinary differential equations (ODEs). In addition, ref. [53] developed a simple mathematical model based on the mass balance principle, and in the same way, ref. [54] in a mathematical model based on Biot's model of consolidation, was also presented, which was expanded with a rheological skeleton for GWL variations. Furthermore, to simulate variations in coastal groundwater flow, researchers [3] employed a process-based numerical model. Similarly, refs [55,56] used MM to determine the primary components behind the water level variation mechanism.

To this end, mathematical models (MMs) can help uncover essential data items and their roles as model inputs by exploring the interplay between variables [57]. In brief, there is also a need for a consolidated literature that summarizes concerns, challenges, and future research objectives in the MM domain.

### 2.3. Combination of Both ML and MM/SM for Groundwater Level Change Prediction

Other researchers have used a combination of ML and MM/SM to forecast changes in groundwater levels. Thus, ref. [58] employed a mixture of statistical and machine learning models, such as entropy-SVM-SG, entropy-SVM-RBF, and entropy-SVM-LN, but did not conduct a systematic review. In another related study, ref. [23] created a sensitivity map to the incidence of land subsidence using statistical and machine learning methods, but this work did not conduct a comprehensive review. Furthermore, ref. [22] employed ANN and MM to estimate groundwater level, although the work was not conducted in a systematic manner. A study by [59] employed extreme learning machine (ELM), modular finite-difference groundwater flow (MODFLOW), and wavelet–extreme learning machine (WA–ELM) methodologies to model groundwater level; again, they failed to perform a systematic review. According to our study, modeling groundwater level fluctuations using a combination of ML and MM methodologies is rare, and new techniques in this domain are needed.

### 2.4. Groundwater Modelling with MODFLOW

According to the United States Geological Survey (USGS), "(MODFLOW) is a finite-difference groundwater flow modeling application that allows you to create a arithmetic representation (i.e., a groundwater model) of the hydrogeologic environment" [60]. MODFLOW can be used to estimate groundwater resources and gain a better knowledge of the system from hydrological, geological, chemical, and hydrological perspectives. A three-dimensional transient groundwater flow model [61] was used to simulate three climate time periods from 1960–1999, 2010–2039, and 2040–2069 to assess the impacts of climate change on groundwater levels. Similarly, MODFLOW-2000 was utilized by [62,63] to better understand regional groundwater flow and Murzuq aquifer systems in Libya, respectively. In addition, ref. [64] used the widely used MODFLOW-2005 model to examine variations in groundwater levels. In addition, ref. [65] used a combined hydrological–hydrogeological model, i.e., employed MODFLOW, under climate change scenarios, defining the spatio-temporal dynamics of water balance and groundwater–surface water (GW–SW) interactions for the upper stream basin of Del Azul. Groundwater numerical models with a traditional rectilinear grid geometry, such as MODFLOW [66], have rarely been used to simulate aquifer test results at a pumping well, in contrast to analytical models, because they are not designed or expected to accurately reproduce the head gradient near the well. MODFLOW is a groundwater modeling approach based on mathematical models. At the time of the present study, the most recent version of MODFLOW 6 was version 6.3.0, which was released on March 4, 2022. Finally, for estimating two-dimensional groundwater flow, ref. [67] used numerical models. According to our research, finding a systematic approach to this problem is difficult.

### 2.5. Related Work

Existing research efforts have sought to review the literature on ML and MM models for GWL predictions, as previously indicated. For example, Tao et al. [68] demonstrated a thorough understanding of the state-of-the-art ML models utilized for GWL modeling as well as the milestones achieved in this domain. Despite their research being beneficial to scholars, it does not answer worries about recent machine learning accomplishments. In comparison to this present study, it is two years behind. Hanoon et al. [69] presented many state-of-the-art artificial intelligence (AI) methods for groundwater quality (GWQ) modeling, as well as a brief description of common AI methodologies. Paepae et al. [70] focused on a high-level overview of critical water quality parameters for a specific use case, as well as the formulation of cost estimates for monitoring them. In a review paper, Saha et al.) [71] reviewed several machine learning and artificial intelligence techniques and methodologies that were commonly used to model and anticipate GWL changes from 2011 to 2020. Other scholars focused their research on measuring model correctness. Ahmadi et al. [72] conducted a systematic review of this subject and evaluated the accuracy of numerous models. According to their statistics, their study was limited to twenty-eight (28) countries. Chiloane et al. [73] examined current development in GWL strategies based on geographic information systems (GIS) and remote sensing. Their research did not pay attention to machine learning model methodologies. In addition, Singh et al. [74] examined an artificial neural network (ANN) model for groundwater level prediction in a study. Their research is not a systematic review, but it does focus on ANN modeling.

In addition, traditional groundwater modeling options, such as certain numerical methodologies, have been proposed [75,76]. Researchers analyzed computing models and simulations for groundwater modeling in a study by Aderemi et al. [77], but they were concerned about existing data collection methods being able to meet computational model criteria and management objectives. Furthermore, in a study by Hussain [78], numerical modeling was used to estimate groundwater levels. They claimed that numerical groundwater modeling is a better alternative to costly aquifer pumping tests for describing aquifer response to external loads. Guevara et al. [79,80] studied three mathematical models to represent variable-density groundwater flow simulations in a systematic investigation.

They were particularly interested in non-Boussinesq effects. The study, however, was insufficiently thorough in meeting the requirements of the systematic review study.

Although the studies mentioned above do not represent the entirety of the literature on machine learning and mathematical models for groundwater level prediction, a search of academic electronic databases for systematic review studies on groundwater level prediction using machine learning and mathematical models yielded few results, if any at all. It is critical to be aware of data on publication trends, popular machine learning algorithms and performance metrics, and mathematical models for groundwater level variations. These data are important for researchers because of the knowledge gap they will fill and the possibility for growth in this field. Despite the fact that systematic reviews may not promise bias-free research, they do reduce bias and give auditable results. As a result of this, it is critical to use a systematic approach to investigate GWL change prediction utilizing machine learning and mathematical models.

### 3. Methodology

This section may be divided by subheadings. It should provide a concise and precise description of the experimental results, their interpretation, as well as the experimental conclusions that can be drawn.

A robust review technique was created to provide a complete and traceable evaluation. For this purpose, review methods serve as a foundation for the review process, reducing researcher bias. The Preferred Reporting Items for Systematic Reviews and Meta-Analyses (PRISMA) flow diagram from Prisma-statement.org [81] was used in this investigation. The information flow across the several phases of a systematic review is depicted in the PRISMA flow diagram. Identifying all possibly relevant data, choosing eligible research, determining bias risk, data extraction, qualitative synthesis of the studies included, and likely meta-analysis [82] are all express and inducible procedures in the systematic review (Figure 1). Drissi et al. [83] exhibited the PRISMA flow diagram in a systematic review, and Pant et al. [84] did the same in a study. The review process was depicted using a diagrammatic presentation.

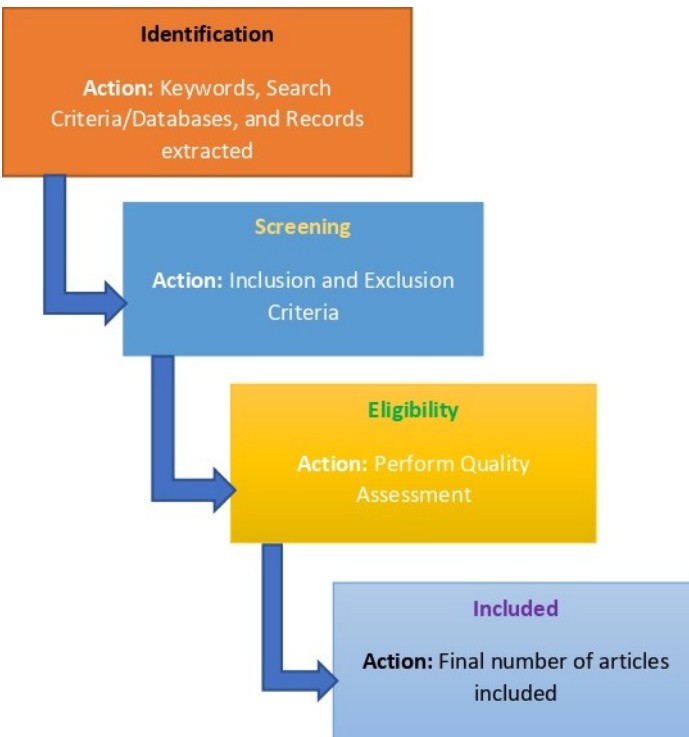

**Figure 1.** PRISMA framework method steps.

Using the PRISMA conceptual framework as a foundation (Figure 1), we created a search string to search the abstract, title, and keywords of literature in the online database "Scopus". The literature sample was compiled from peer-reviewed journal articles published in English between January 2000 and May 2022. On 15 May 2022, the search process was completed.

The procedure went through four steps (Figure 2) before obtaining the 117 most important studies on groundwater level change modeling using machine learning and mathematical model techniques.

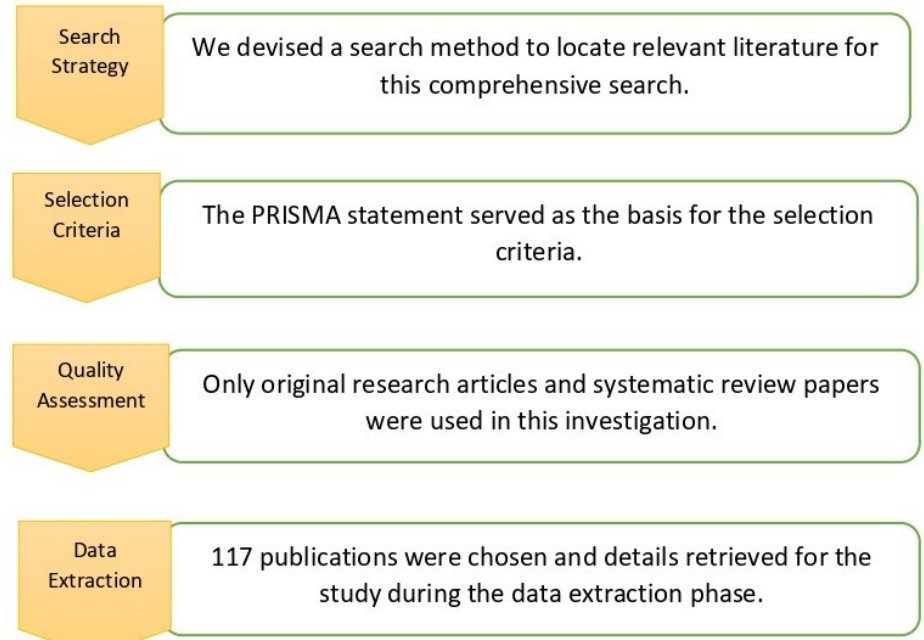

**Figure 2.** Steps of the systematic literature review's analysis process.

The search string was designed to suit specific exact keywords, such as "groundwater", "mathematical models", "groundwater flow", "groundwater resources", "climate change", "computer simulation", "numerical model", "statistical model", "machine learning", and "groundwater level changes", through a well-thought-out search strategy. Many articles were found using the search query, but many were eliminated during the title and abstract screening phase. The search string appears in the text field (Table 1) below.

**Table 1.** The search query.

| The Search String |
|---|
| TITLE-ABS-KEY((MACHINE LEARNING AND GROUNDWATER LEVEL CHANGES) OR (MATHEMATICAL MODELS AND GROUNDWATER LEVEL CHANGES)) AND (LIMIT-TO (PUBYEAR, 2022) OR LIMIT-TO (PUBYEAR, 2021) OR LIMIT-TO (PUBYEAR, 2020) OR LIMIT-TO (PUBYEAR, 2019) OR LIMIT-TO (PUBYEAR, 2018) OR LIMIT-TO (PUBYEAR, 2017) OR LIMIT-TO (PUBYEAR, 2016) OR LIMIT-TO (PUBYEAR, 2015) OR LIMIT-TO (PUBYEAR, 2014) OR LIMIT-TO (PUBYEAR, 2013) OR LIMIT-TO (PUBYEAR, 2012) OR LIMIT-TO (PUBYEAR, 2011) OR LIMIT-TO (PUBYEAR, 2010) OR LIMIT-TO (PUBYEAR, 2009) OR LIMIT-TO (PUBYEAR, 2008) OR LIMIT-TO (PUBYEAR, 2007) OR LIMIT-TO (PUBYEAR, 2006) OR LIMIT-TO (PUBYEAR, 2005) OR LIMIT-TO (PUBYEAR, 2004) OR LIMIT-TO (PUBYEAR, 2003) OR LIMIT-TO (PUBYEAR, 2002) OR LIMIT-TO (PUBYEAR, 2001) OR LIMIT-TO (PUBYEAR, 2000)) AND (LIMIT-TO (DOCTYPE, "ar")) AND (LIMIT-TO (EXACTKEYWORD, "Groundwater") OR LIMIT-TO (EXACTKEYWORD, "Mathematical Models") OR LIMIT-TO (EXACTKEYWORD, "Groundwater Flow") OR LIMIT-TO (EXACTKEYWORD, "Groundwater Resources") OR LIMIT-TO (EXACTKEYWORD, "Climate Change") OR LIMIT-TO (EXACTKEYWORD, "Computer Simulation") OR LIMIT-TO (EXACTKEYWORD, "Numerical Model") OR LIMIT-TO (EXACTKEYWORD, "Ground Water")) AND (LIMIT-TO (LANGUAGE, "English")) AND (LIMIT-TO (SRCTYPE, "j")) |

There were 237 articles found that matched the search-string criteria. The title, keyword, abstract, digital object identifier (DOI), and other information from the articles were recorded in a comma-separated value (CSV) file (generated from the Scopus database). The articles were saved in a reference manager software (Mendeley). According to Mendeley.com accessed on 1 June 2022, "Mendeley Reference Manager is a free web and desktop reference management tool. It aids you simplify your reference management workflow so you can focus on achieving your goals" [85,86].

Review questions were constructed during the design stage to elicit the study aims, which subsequently formed the foundation of the inquiry, as indicated by Brereton et al. [87]. The approach of the Goal-Question-Metric (Table 2) [88] was utilized. The technique has been shown to be effective for obtaining systematic review objectives by Lun et al. [89] and Wiafe et al. [90].

**Table 2.** Goal-Question-Metric espoused from [88].

| Purpose | This research examines. |
|---|---|
| Issues | Trends in publishing, application domains, methodologies, and future directions. |
| Object | Prediction of groundwater level variations with existing machine learning and mathematical model approaches. |
| Viewpoint | Between the years 2000 and 2022. |

Within the domain of groundwater level changes, issues about the most often used ML and MM methodologies and their performances, as well as present and future research directions, raise questions. As a result, the goal of this research is to look into existing studies in the field in order to fill in these knowledge gaps. Questions about the review process were raised and answered, as well as the logic or motivation for doing so (Table 3).

**Table 3.** Review interrogations and motivation.

| | Research Question | Inspiration | Technique |
|---|---|---|---|
| RQ1 | What are the publication trends in groundwater level changes using ML and MM approaches? | To categorize studies and evaluate their foci, dominant venues, and contributions. Trends can be examined throughout time depending on the quantity of studies. These data will show the scientific community where there are gaps in the domain. | Quantitative |
| RQ2 | In groundwater level change modeling, which ML and MM approaches are used? | To identify the several ML and MM techniques that are currently being utilized to predict groundwater level changes. This will reveal which method is the most popular. It will also reveal why it is the most popular option. Researchers will be able to see the potentials and/or lack of concentration in various ML and MM methodologies based on this information. | Quantitative |
| RQ3 | What effect have these ML and MM techniques had on groundwater level changes? | To determine the present impact of ML and MM approaches on groundwater level changes, as well as to study and categorize existing ML and MM approaches for groundwater level changes based on the specific challenges they attempt to solve. | Qualitative |
| RQ4 | What will be the research focus in the future? | This topic aims to suggest future ambitions for researchers and practitioners in using ML and MM methodologies to predict groundwater level changes. It provides newbie researchers with information on current subjects of interest in the domain. | Qualitative |

Other publications were found through a search on Google Scholar and Google, based on our study's topic and objectives. From Scopus, Google Scholar, and Google search, there were a substantial number of duplicate articles. The problem of duplication was solved using Mendeley's automatic duplicate elimination procedure. We examined if the title, keywords, and DOI of the records in the CSV file were the same, and eliminated the duplicates accordingly to ensure the process's credibility.

After removing duplicates, the screening procedure began with 410 records, which included the inclusion and exclusion criteria. Two reviewers worked simultaneously and independently to qualify the titles of articles that would be kept or eliminated. In some cases, the reviewers submitted article titles for consideration to third-party reviewers who are experts in the field. The agreement was the final judgment if both critics agreed to maintain or remove a certain record. In the end, 290 reports were assessed for eligibility (Figure 3). The papers were randomly assigned to five reviewers from two different higher education institutions. Each reviewer evaluated the allocated records to see if they fulfilled the exclusion and inclusion criteria. The exclusion and inclusion criteria were used to keep or remove papers that were more relevant to our topic.

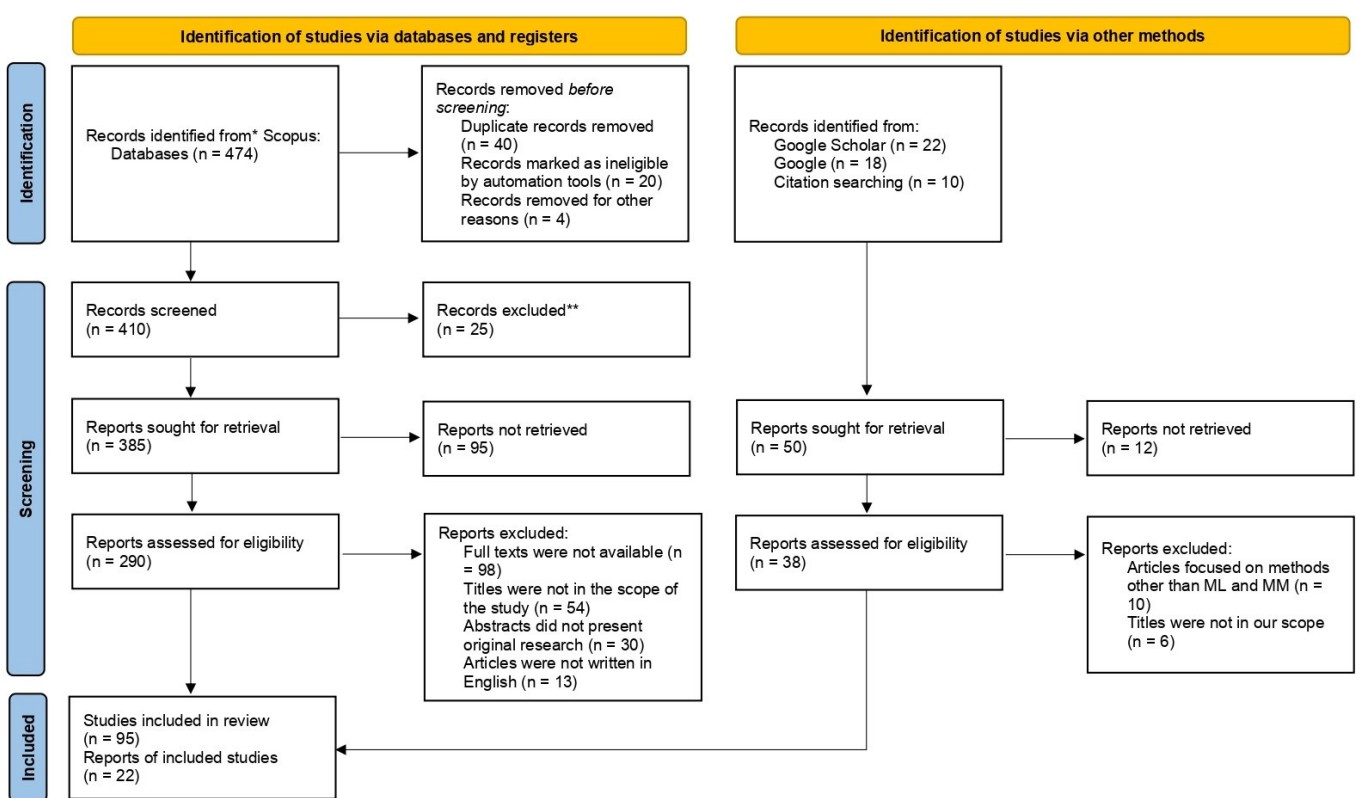

**Figure 3.** The PRISMA diagram for inclusion and exclusion criteria of literature. * means all the records were from Scopus, whiles ** means the records were discarded in its entirety.

For this study, a report may have been removed (a) or met (b) the criteria on:

a.  Exclusion criteria: Our research disqualifies any report that meets the following criteria:
   - There were no full texts available. As a result, 98 reports were deemed unfit;
   - Titles were not in the scope of our study. As a result, 54 reports were discarded;
   - Original research was not presented in the abstracts. As a result, 30 reports were deleted;
   - The articles were written in non-English language. As a result, 13 reports were removed.

b.  Inclusion criteria: Reports that matched the following criteria were included in our research:

- The articles are published in English, and the abstracts contain original research on the topics;
- Articles in our scope introduce machine learning and mathematical model techniques.

We excluded conference, short paper, and workshop publications from the scope of this study. Following the reviewers' full-text screening, 195 reports were deleted from the exclusion criteria. At the completion of the systematic review, there were 117 papers (Figure 3): 95 studies included in the review, and 22 reports of included studies from Google Scholar, Google, and citation searching (Figure 3).

Finally, another reviewer double-checked the included papers to guarantee that the data had been mined correctly. All members of the research team were assigned to serve as second reviewers, confirming that the other reviewers' extracted and included primary studies (117 articles) were accurate and answered the research questions in Table 3. The primary studies (PS) used in this review are listed in Appendix A.

## 4. Results and Discussion

The year of publishing, the address and affiliation of the accompanying author, and the journal outlet (publisher) were all noted. In addition, the publication ID, DOI, and authors' PubMed ID were noted for each piece. The ML and MM approaches that were used are also discussed. Studies that looked at how to improve existing methods were also analyzed and summarized, as well as the analysis of the graphic maps. The following is a discussion of the study's findings.

### 4.1. Publication Trends

Over time, the quantity of research articles on ML and MM techniques for groundwater level change prediction has expanded. Between 2000 and 2015, the use of ML and MM approaches for predicting groundwater level changes increased and decreased within the same period (Figure 4), but it began to gain traction again after 2016. From 2000 to 2016, thirty-seven articles accounted for 31.62 percent of all primary research reviewed. There were no papers published in 2001, 2005, or 2013. Based on the study's goal, there was a surge in 2021, which represents 28 (23.93 percent) of the 117 reviewed papers on ML and MM techniques. Since 2016, publications in the domain saw an exponential increase up until 2021, i.e., the steepest slope. In 2019, the number of publications in the area climbed from six to sixteen, increased by one in 2020, and increased by eleven in 2021. As of the time the survey was conducted in 2022, nine articles had been recorded.

Based on the findings, the primary articles (117 articles) were also slanted towards publishing firms/outlets. Out of the primary articles, Elsevier BV had the most papers with 30 (25.64%), followed by Multidisciplinary Digital Publishing Institute (MDPI) with 20 (17.09%), Springer Science and Business with 14 (11.97%), and Blackwell Publishing Limited with 10 (8.55%). These four publishers published 63.25 percent (74) of the 117 primary articles (Figure 5). The fewest number of articles recorded by a publishing house was one. The remaining publishing houses only have eight or fewer studies.

We discovered that the primary studies (117 articles) are skewed by country in this systematic review. China had the most corresponding authors with thirty-eight, followed by the United States (US) with twenty-five, and the United Kingdom (UK) with nine (Figure 6). According to the associated writers' addresses, the majority of the papers originated in Asian countries. Asia was responsible for 56 of the 117 articles (47.9 percent). North America had twenty-five articles, while South America had three. Africa was represented by two articles: from South Africa and Egypt. Figure 6 summarizes the distribution of publications based on geographic location.

The proportion of the top five countries is also shown (Figure 7). China accounted for 32% of all primary studies, the US for 21%, the UK for 8%, Iran for 4%, and Portugal, Japan, and Spain for 3% each.

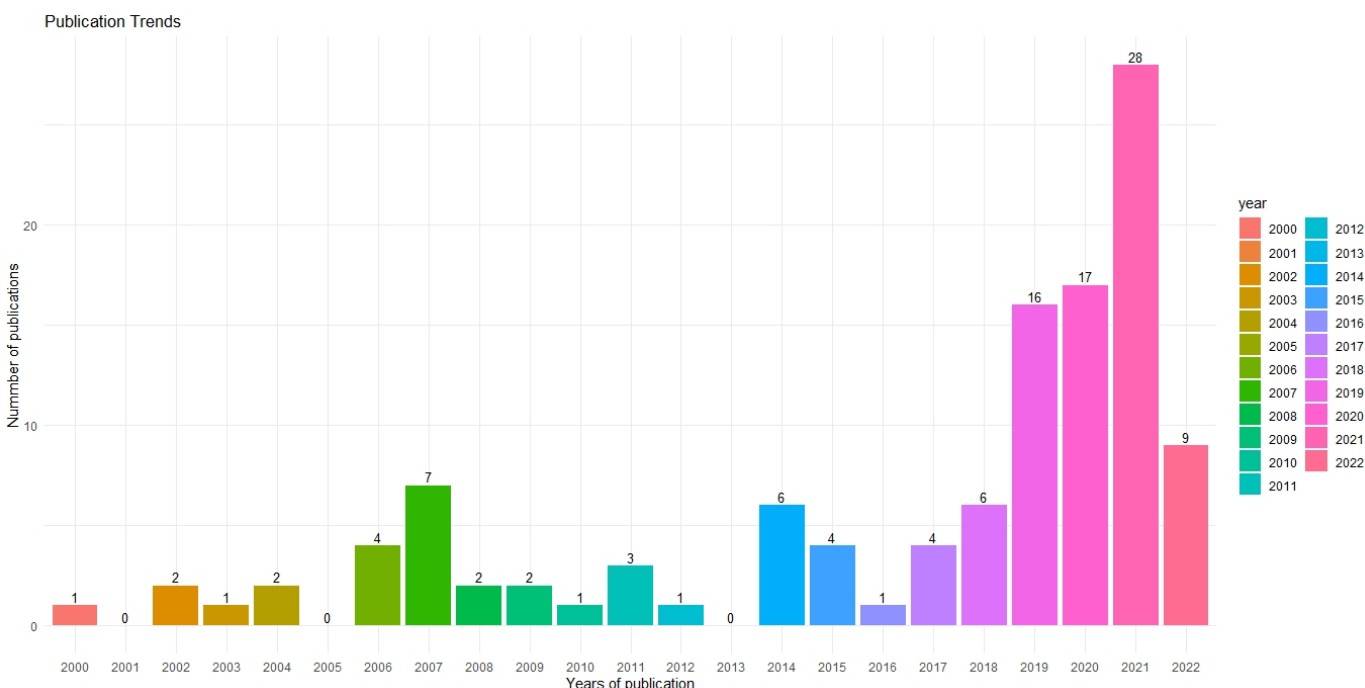

**Figure 4.** Publication trends in primary articles from 2000 to 2022.

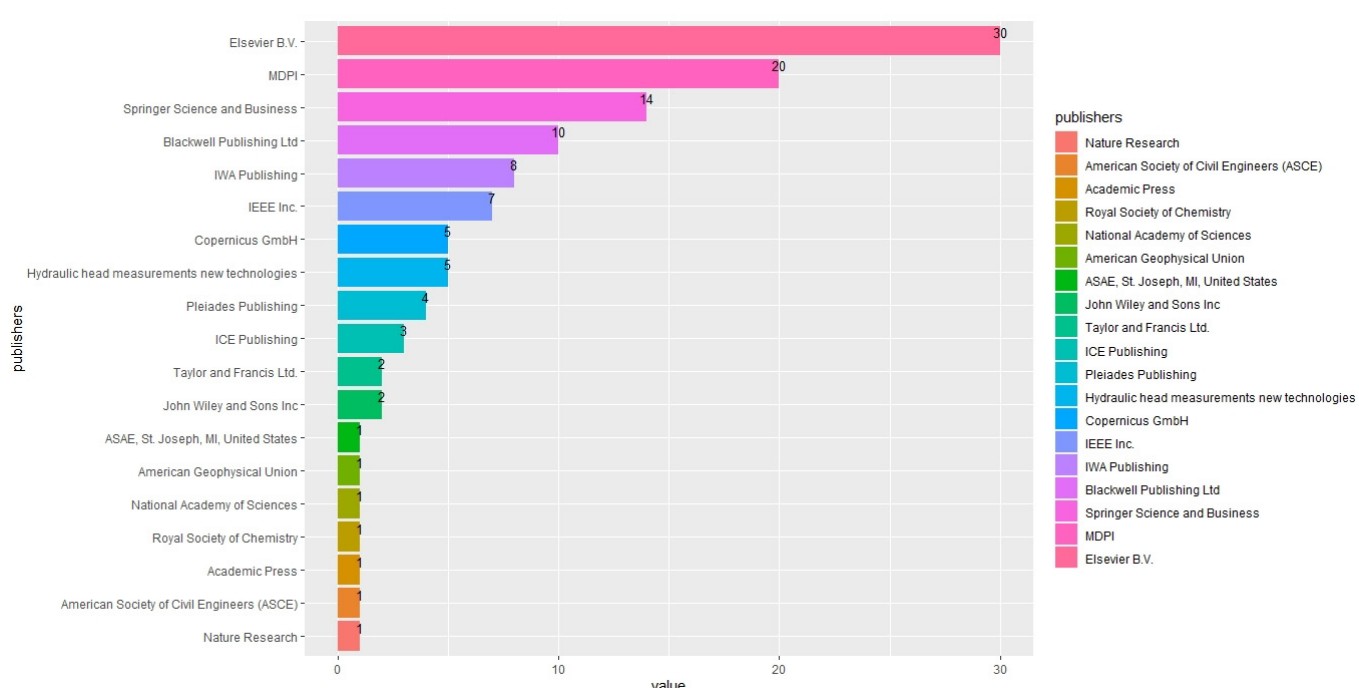

**Figure 5.** Publication outlets from the primary studies.

Predictor Variables for Modeling GWL Changes

The primary articles used in this study used input data to predict groundwater level changes and address their impact, and our study will look at the climatic data from those studies as well. According to NASA.gov accessed on 9 June 2022 [91], "climate change is a shift in the usual weather found in a place". Groundwater level changes are influenced by climate conditions [92], and climate change is threatening our future. Extreme occurrences such as droughts and floods, as well as rising precipitation variability due to climate change, have a severe impact on groundwater quantity and quality. Precipitation, temperature,

rainfall, Gravity Recovery and Climate Experiment (GRACE) data, ENVISAT advanced synthetic aperture radar (ASAR) satellite data, and lithology are among the many predictor variables. The primary studies' predictor factors are provided (see Table 4).

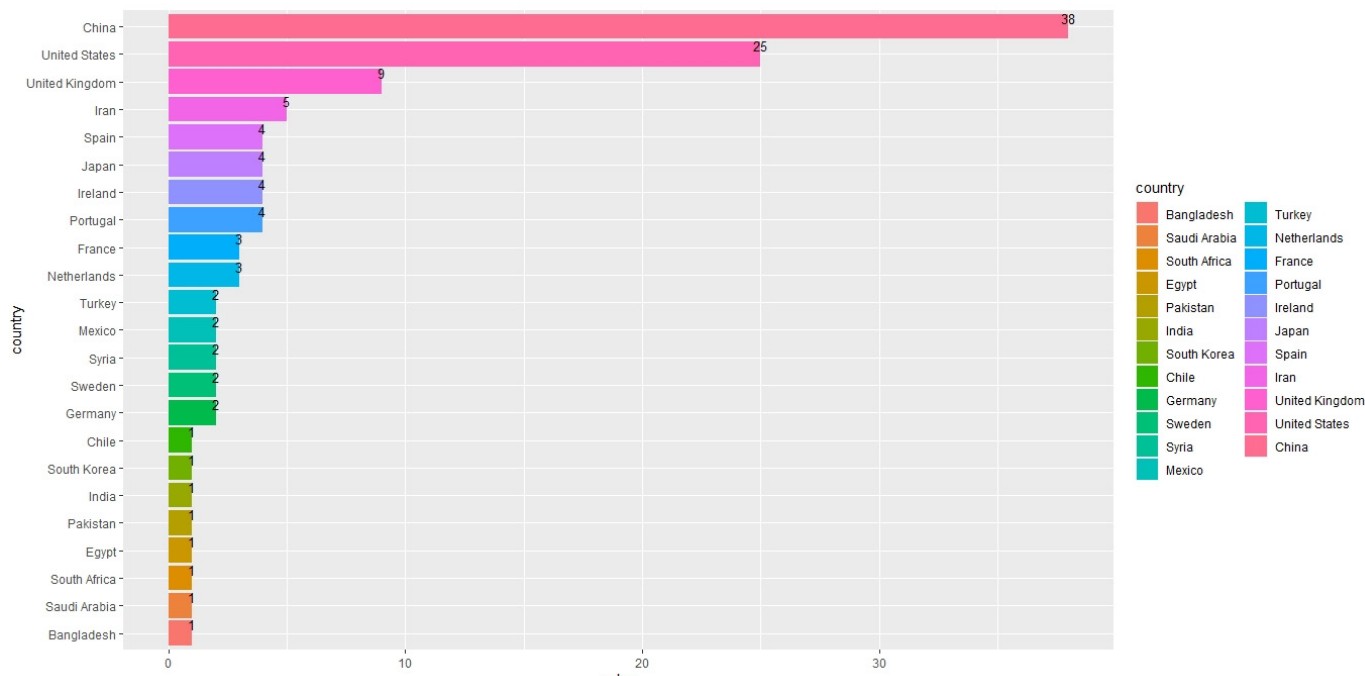

**Figure 6.** Distribution by country, based on origin of corresponding author.

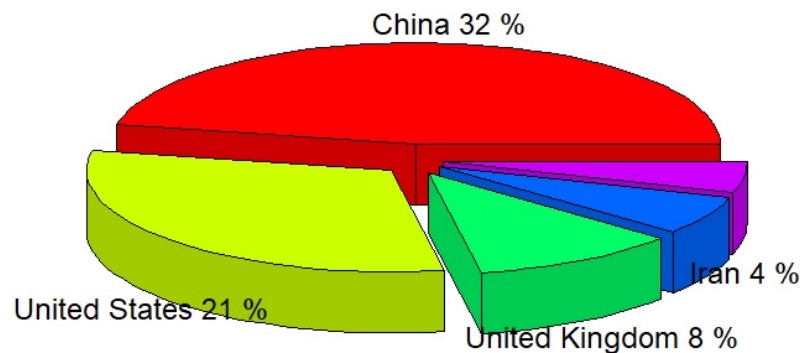

**Figure 7.** Proportion of top five countries from the primary studies.

The groundwater level changes were predicted with the predictor factors from our primary investigations (117 articles) (Table 4). Precipitation was used in many of the studies. According to education.nationalgeographic.org accessed on 9 june 2022 [93], any liquid or frozen water that forms in the atmosphere and falls back to Earth is known as precipitation. Rain, sleet, and snow are only a few examples.

*4.2. Techniques in Modeling GWL Changes*

Groundwater level changes over the years have been predicted with ML [2] and MM [56] approaches. This section examines how both strategies have been employed in research. According to our data (main studies), ML techniques accounted for 45.3 percent (53) of the publications, MM approaches for 39.3 percent (46) of the articles, and 5.1 percent (6) were cases where both ML and MM approaches were used in a study. The MODFLOW technique, which aids in the accurate, dependable, and efficient simulation of groundwater flow, accounted for 10.3% (12) of the primary publications (Figure 8).

**Table 4.** Common predictor variables from the primary study.

| Predictor Variable | Type |
|---|---|
| Precipitation | Continuous |
| Temperature | Continuous |
| Streamflow | Continuous |
| Irrigation demand | Continuous |
| Rainfall | Continuous |
| Population growth | Categorical |
| Environmental regulations | Categorical |
| Land elevation | Categorical |
| Lithology | Continuous |
| Distance from river | Continuous |
| Plan and profile curvature | Categorical |
| Land use | Categorical |
| Sea level | Continuous |
| Evapotranspiration | Continuous |
| Soil moisture | Categorical |
| Humidity | Continuous |
| Canopy water | Continuous |
| Vegetation | Categorical |
| Sunshine | Continuous |
| Average wind | Continuous |
| GRACE data | Satellite data |
| ENVISAT ASAR (EA) | Satellite data |
| RADAR SAT-2 | Satellite data |

### 4.2.1. ML Approaches and Performance Metrics

Machine learning techniques have been used to forecast changes in groundwater levels and accounted for 45.3 percent (53) of the publications (Figure 8). The many papers that were included in the review yielded over 70 different algorithms (Table 5). Random forest (RF), artificial neural network (ANN), multilayer perceptron (MLP), support vector machine (SVM), long short-term memory (LSTM), adaptive boosting (AdaBoost), extreme gradient boosting (XGBoost), recurrent neural network (RNN), adaptive neuro-fuzzy systems (ANFIS), and metaheuristic algorithms such as differential evolution (DE) and the genetic algorithm (GA) were the dominating algorithms. Other ensembles of the aforementioned algorithms, such as extreme learning machine (ELM) crossbred with ant bee colony (ABC), improved grey wolf optimizer (IGWO), and whale optimization algorithm (WOA), were used. Singular spectrum analysis (SSA), support vector regression and support vector classification (SVR-SVM) were the other ensembles.

The most commonly utilized algorithm was RF (20 studies), which was followed by ANN (15 studies). AdaBoost and XGBoost were employed in 9 and 8 studies, respectively, while SVM was utilized in 13 studies. In eight (8) and five (5) studies, respectively, ensembles and metaheuristic algorithms were applied. In 20 studies, various algorithms such as decision tree (DT), convolutional neural network (CNN), ZeroR, Fuzzy Logic (FL), and others were utilized. It should be noted that one article in the primary study can employ three or more ML approaches. Figure 9 summarizes the number of times ML approaches were employed in studies.

In the primary research, statistical indicators were utilized to assess the accuracy of the ML models used to forecast groundwater level changes. According to our statistics, the root mean square error (RMSE) is the most widely utilized performance indicator for monitoring accuracies, accounting for 30.8 percent. Nineteen evaluation metrics were identified in the study. Table 6 shows the number of times performance metrics were employed and their proportions. The Nash–Sutcliffe efficiency coefficient (NSE), mean absolute error (MAE), coefficient of correlation (R), coefficient of determination ($R^2$), percentage bias (PBias), mean square error (MSE), mean absolute percentage error (MAPE), and Pearson coefficient (PR) were also used to measure the accuracy of the models. Other metrics included Monte Carlo,

normalized mean square error (NMSE), Shannon entropy, area under the curve (AUC), Akaike's information criteria (AIC), average squared error (ASE), area under the receiver operator characteristics curve (AUROC), Kling–Gupta efficiency (KGE), scatter index (SI), and mean absolute relative error (MARE).

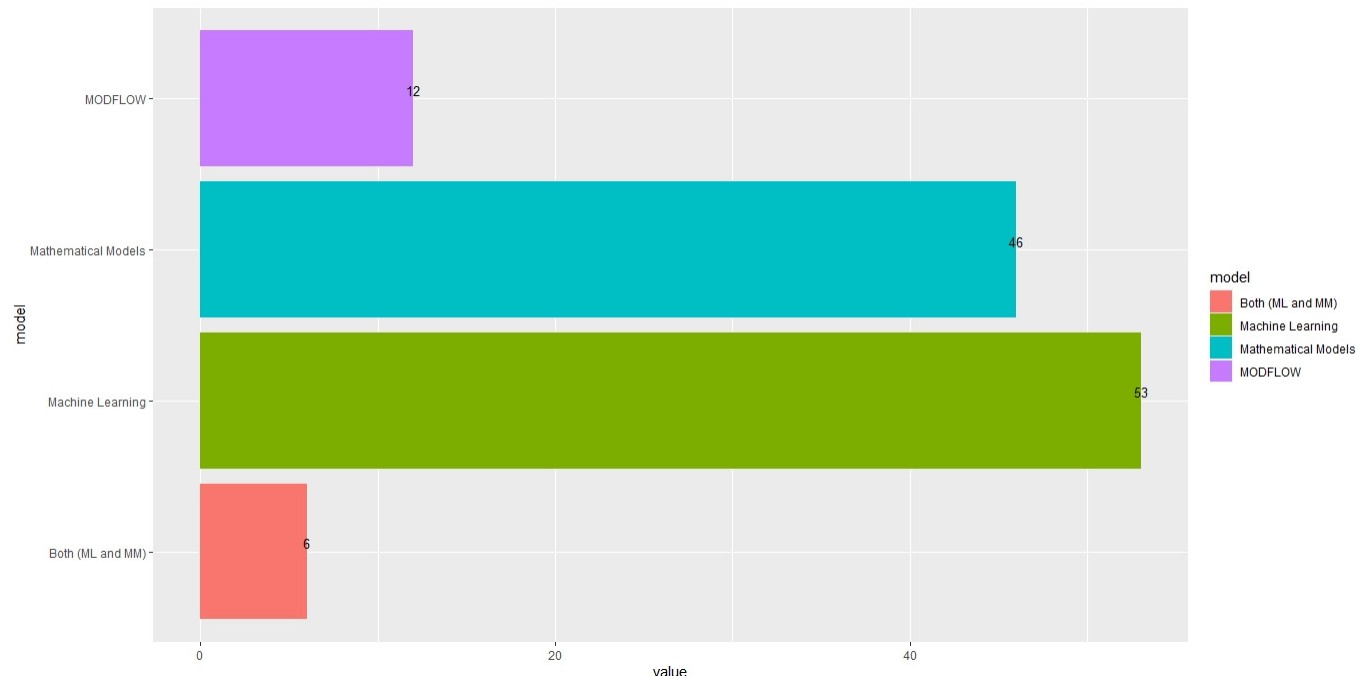

**Figure 8.** Techniques for modeling groundwater flows.

**Table 5.** ML techniques for predicting GWL changes.

| Algorithm | Number of Times Used in Primary Study |
|---|---|
| RF | 20 |
| SVM | 13 |
| ANN | 15 |
| LSTM | 7 |
| MLP | 8 |
| AdaBoost | 9 |
| XGBoost | 8 |
| ANFIS | 5 |
| RNN | 4 |
| Ensembles | 8 |
| Metaheuristic Algorithms | 5 |
| Others | 20 |
| Total | 122 |

NMSE, RMSE, Monte Carlo, Shannon entropy, AUC, and MAPE were discovered to be the most often used performance measures in the ML ensembles approach. The Pearson coefficient (PR) was also observed being used in deep learning algorithms. KGE, NSE, and PBias were also used to evaluate XGBoost machine learning algorithms. MAE, RMSE, and $R^2$ performance metrics were also used in hybrid ML models. The root mean square error (RMSE) represents the model's absolute fit to the data and is an acceptable degree of performance with the same units as the projected. The coefficient of determination ($R^2$), on the other hand, is a relative statistic that does not represent the model's absolute correctness. Furthermore, Monte Carlo is utilized to measure forecast uncertainty.

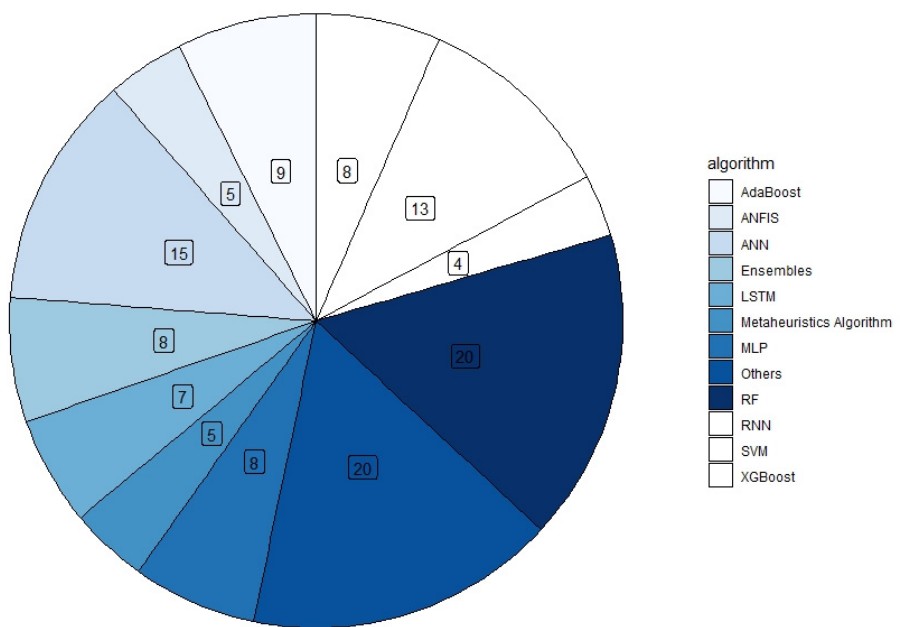

**Figure 9.** Number of times ML approaches were employed in studies.

**Table 6.** Performance metrics and their proportions.

| Performance Metric | Number of Times Employed | Percentage (%) |
|---|---|---|
| RMSE | 32 | 30.8 |
| MSE | 7 | 6.73 |
| MAE | 11 | 10.6 |
| NSE | 11 | 10.6 |
| MAPE | 4 | 3.8 |
| R | 11 | 10.6 |
| Monte Carlo | 2 | 1.9 |
| NMSE | 1 | 0.96 |
| Shannon Entropy | 1 | 0.96 |
| AUC | 1 | 0.96 |
| AIC | 1 | 0.96 |
| $R^2$ | 11 | 10.6 |
| PBias | 4 | 3.83 |
| PR | 2 | 1.9 |
| ASE | 1 | 0.96 |
| AUROC | 1 | 0.96 |
| KGE | 1 | 0.96 |
| SI | 1 | 0.96 |
| MARE | 1 | 0.96 |
| Total | 104 | 100 |

### 4.2.2. MM Approaches and Performance Metrics

Mathematical and statistical models accounted for 39.32 percent (46 papers) of the primary investigations (117 articles) used (Figure 8). The most prevalent approaches were the Boussinesq equation, Biot's model, rheological model, cross-correlation analysis, Laplace transform (LT), Laplace homotopy perturbation model (LHPM), and Bayesian model averaging. The Glover–Dumm mathematical model was also used in primary investigations that took into account MM methods. The Glover–Dumm MM technique, on the other hand, was coupled with other models such as frequency ratio (FR) by the researchers to produce a proposed model. The most widely used performance indicators are the coefficient of determination (CD), the modeling efficiency (EF), the maximal error (EF), coefficient of residual mass (CRM), and the root mean square error (RMSE). The MM

techniques and assessment criteria utilized to estimate groundwater level changes derived from our primary investigations are shown in Table 7.

**Table 7.** MM approaches and performance metrics.

| MM Approaches | Performance Metrics |
|---|---|
| LT | RMSE, EF |
| LHPM | CRM, CD |
| Boussinesq equation | CD, CRM, EF |
| Bayesian modeling | Posterior, predictive checks, AIC, RMSE |
| Rheological model | EF, RMSE |
| Glover–Dumm model | CRM, R, $R^2$ |
| Proposed Models | RMSE, MAE, AIC, etc. |

The majority of the MM techniques were clearly deployed between the years 2000 and 2011. Between 2012 and 2022, few MM methods were used.

### 4.3. Impacts of ML and MM Approaches on GWL Changes

When ML is compared to mathematical model techniques, the application of machine learning approaches for groundwater level changes has increased throughout the years. The application of single machine learning algorithms and the presentation of novel methods are the most significant contributions of ML and MM approaches in predicting groundwater level variations. Due to the difficulty of computational complexity in machine learning algorithms [94], research towards lowering computational times is required and significant. Although computational complexity can impair application performance, several studies have found that performance times are reduced. As utilized in studies by Jiang et al. [2] and Kayhomayoon et al. [95], the employment of some ensembles of machine learning and metaheuristic algorithms helps to minimize computational time and enhances the prediction of groundwater level changes.

Furthermore, model performance and computing efficiency are critical, and mathematical models are not excluded. Our primary research revealed that MM techniques have not been applied in a long time, resulting in a gap. However, it is important to recognize their significance in anticipating groundwater level variations. Researchers said in a study by Li et al. [96] that "there is no viable mathematical approach and theory to be utilized for modeling and predicting the fluctuation in the permafrost groundwater", but they did not rule out the use of mathematical models in predicting groundwater level variations.

This research has added to our understanding of the machine learning and mathematical models commonly used in predicting groundwater level changes, revealing that machine learning models are used more frequently than mathematical models, with the former providing more effective ways to predict groundwater level changes. Also, SVM [18], RF [38], ensembles of ML [36], and some statistical techniques [13] can increase accuracies, computational efficiency, and performance in groundwater level change prediction. Jyolsna et al. [11] also proved that using imagery to predict groundwater level fluctuations improves prediction performance.

Currently, the application of machine learning ensembles has been shown to produce higher accuracies in predicting groundwater level fluctuations. For example, Yadav et al. [36] showed that ensembles improve prediction accuracy, achieving an accuracy of 89.3 percent, which is the current attained accuracy for ensembles based on our core articles (117 articles). The improvement of planning and management of water resources depends on the development of an accurate soft computing method for groundwater level (GWL) forecasting [97], and this study has brought great news to the literature by revealing that random forest (RF) [10], support vector machine (SVM) [36], and artificial neural network (ANN) [9] methodologies are widely used machine learning algorithms for modeling groundwater level changes. Additionally, the most often employed GWL predictor factors were identified by this study (Table 4). Our research has shown that machine learning, including its

hybrid [7] and ensemble [8] models, operates flawlessly when employing satellite images by increasing computing complexity and processing time.

### 4.4. Future Directions of ML and MM Approaches in Predicting GWL Changes

Although the use of machine learning and mathematical model techniques has expanded since 2016, machine learning accounts for the majority of primary publications. Between 2000 and 2009, the majority of the MM techniques were employed to forecast groundwater level fluctuations. Global climate change has already had visible consequences on the environment, and groundwater, which serves as a supply of drinking water for humans, remains a key worry. Challenges and issues of climate change continue to evolve. Today, the world strives for technology, which has a massive impact. Machine learning and mathematical model techniques are making life easier since they learn from previous events and forecast future ones. Researchers employed machine learning algorithms more frequently than mathematical models in the primary publications included in this systematic review. Figure 8 shows that ML accounted for 53 of the 117 articles, whereas MM accounted for 46. It has been discovered that MM approaches for predicting groundwater level changes have been used less frequently in recent years. Machine learning has surpassed mathematical models, and the random forest (RF) algorithm is the most commonly used algorithm for predicting groundwater level fluctuations, according to Table 3. Some algorithms in Table 3 were classed as "other", and they appeared 20 times in the sampled data. These algorithms were found to be utilized in fewer than three articles. Decision tree (DT) and multiple linear regression (MLR) are examples of the "other" algorithms. To predict groundwater level changes, researchers are turning to newer, maybe more inventive methodologies. As a result, future research may employ unique methodologies such as hybrid and ensemble models. Furthermore, researchers may again use hybrid mathematical models, combining new models with old mathematical models. Nonetheless, it should be noted that academics are worried about the requirement for models to minimize computational complexity, to be computationally efficient, and increase performance accuracy. Furthermore, based on some findings in our main research, increasing activism and education on the effects of climate change are essential.

### 4.5. Analysis of Graphic Maps

This part uses the VOSviewer software [98] to give an examination of scientific graphical maps based on bibliographic data. Bibliometrics is a branch of research that examines citations and keywords in the context of scholarly publishing. Scholars can use such techniques to better comprehend the topography of scholarly debate as it appears in the literature.

#### 4.5.1. Co-citation of Authors

The goal of the author co-citation analysis [99] is to reveal the structure and linkages between the writers who are most frequently mentioned together [100]. Figure 10 shows a bibliometric map with five subject clusters formed by co-citation links between authors. Rodell is an influential author with the greatest number of citations (58 citations), as can be seen from the second node (Cluster 2: green color).

#### 4.5.2. Co-Citation of Journals

Once two articles published in different journals acquire a citation from a third paper published in another journal, this is known as journal co-citation [101]. The co-citation of journals from our primary study is depicted in Figure 11. The larger the node, the more documents that have been published. We can see that there are three groups, with Journal of Hydrology being the most referenced of them all.

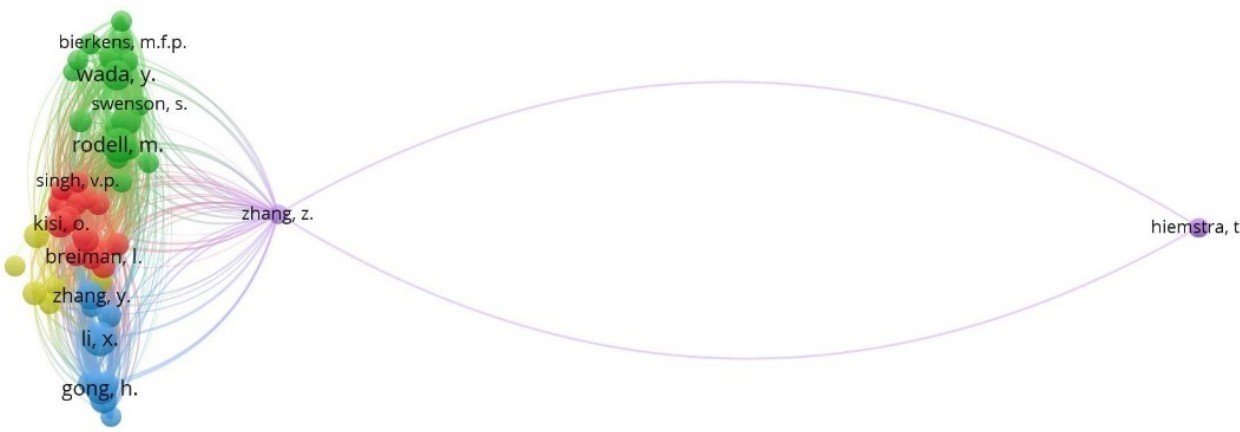

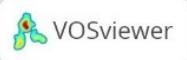

**Figure 10.** Co-citation of authors. The colors represent clusters.

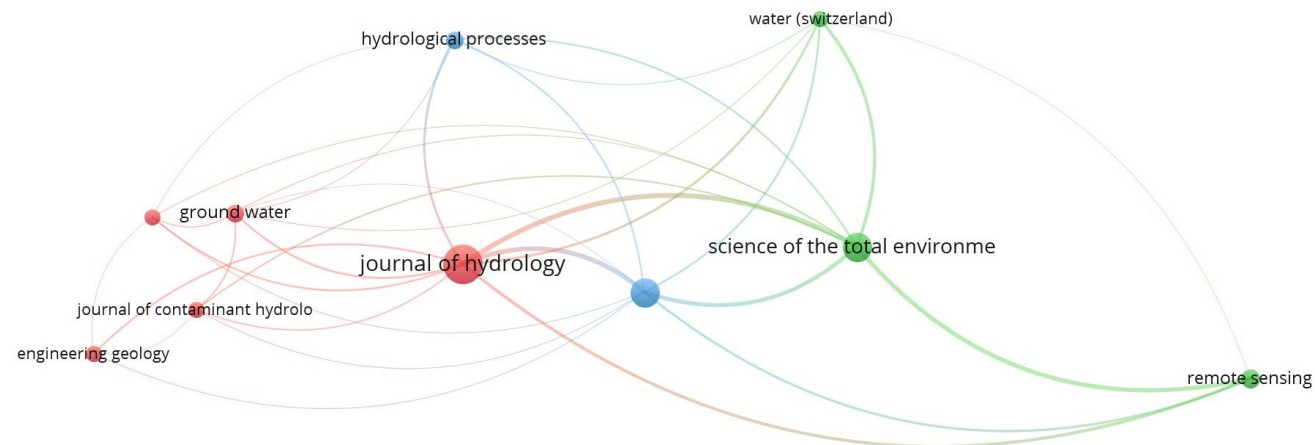

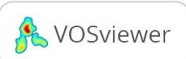

**Figure 11.** Co-citation of journals. The colors are clusters. Here we have three clusters.

### 4.5.3. Bibliographic Coupling by Countries

The graphical map of bibliographic connection among the major countries is shown in Figure 12. It displays fascinating interconnections between them. China, Iran, Australia, and Italy were bibliographically linked to the United States. However, based on addresses of the corresponding authors from the primary studies (117 articles), we discovered that many of the articles were from China.

### 4.5.4. Bibliographic Coupling of Author Keywords

Figure 13 depicts the author keywords layered by their average year of publication, with colors indicating temporal variability. A content analysis can be carried out to provide

quantitative measures by gathering keywords. When it comes to exploring new fields, this strategy looks promising.

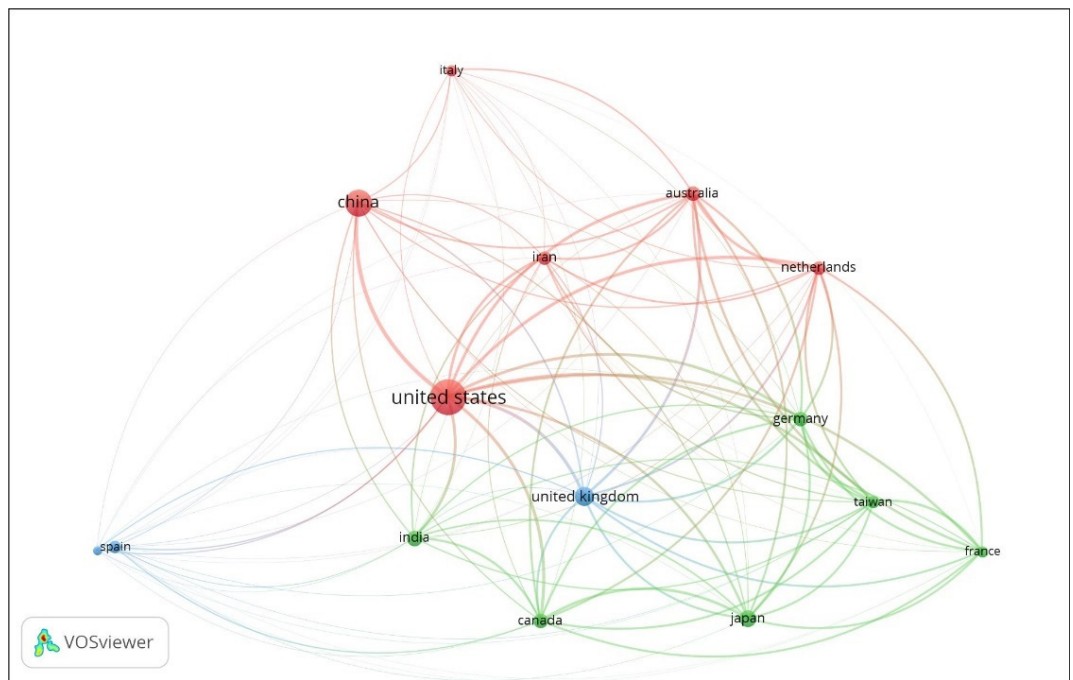

**Figure 12.** Bibliographic coupling by countries.

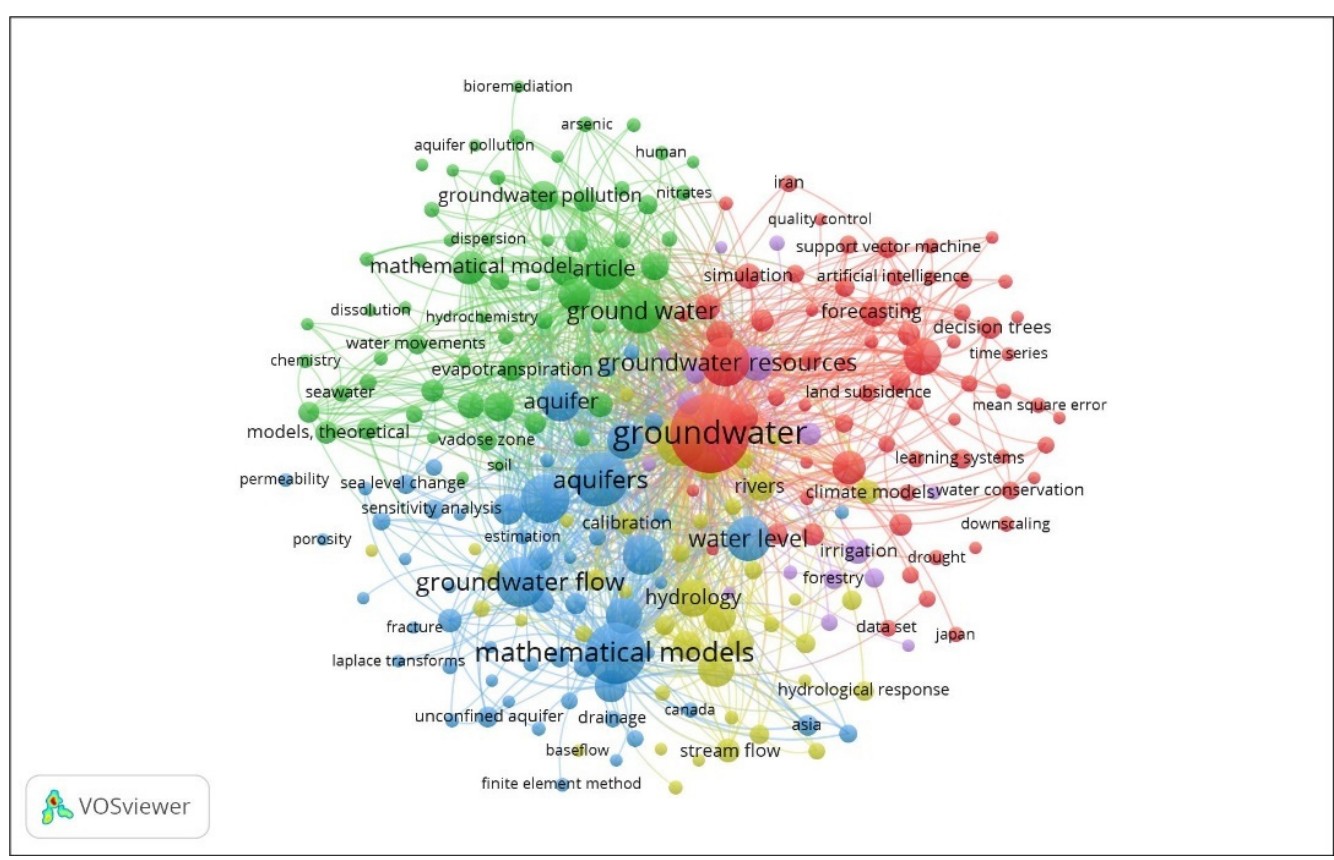

**Figure 13.** Co-occurrence of author keywords with overlay visualization.

## 5. Research Legitimacy, Limitations, and Conclusions

This article presents an overview of existing research on the application of machine learning and mathematical model techniques for modeling groundwater level changes. It is one of the few systematic reviews that we are aware of, if not the first, that considers both machine learning and mathematical model approaches in literature, and also considers their impact on groundwater level changes. However, it is the first in a systematic review to analyze ML and MM techniques in predicting groundwater level changes.

Methods validation in research is critical, so we must validate our approach. The data for this study were gathered using the Scopus database. Possible paper omissions during the selection process, as well as data extraction bias (consideration of papers from only the Scopus database), are threats to the study's validity. Considering only the Scopus database, there are a number of other databases, including IEEE, Web of Science, and ACM, that may contain related studies.

Our findings in this study cannot be generalized, but they do show how ML and MM techniques have been utilized in the domain to forecast groundwater level changes. Furthermore, in order to reduce bias in paper selection, the PRISMA [81] guidelines were used, which have been shown to be effective, as shown in a work by Alfadil et al. [82]. Multiple people carefully selected the search string, database, and scope. Furthermore, the protocol was reviewed by domain experts as external reviewers to eliminate any potential bias. The groundwater search keywords were carefully chosen to ensure a descriptive scope and to extract papers for the review.

The study only considered journal articles to ensure that the primary studies were of excellent quality and have undergone peer review; while journal articles are not without bias, ensuring quality is critical. Although the omission of conference papers may introduce some bias, it is difficult to select the best ones from the vast number of available proceedings. Researchers assessed the paper quality on conference proceedings at their discretion, which would have added an additional bias, hence the omission of conference papers in this study.

In fact, the study demonstrated that various researchers have used ML and MM approaches to predict groundwater level changes. Although both ML and MM techniques are used, according to our primary articles (117 articles), the former is mostly used for the same purpose in recent years, presenting a significant skewness in the domain. The state of the water as well as its level are interesting aspects of groundwater modeling (e.g., concentrations of different chemicals and geospatial analysis).

Machine learning has lowered computational complexity and shortened model training times. Furthermore, researchers frequently used the support vector machine (SVM), random forest (RF), and artificial neural network (ANN) algorithms in their studies, with little emphasis on other algorithms such as ensembles, hybrid, and metaheuristic algorithms. Researchers are encouraged to use mathematical model approaches to predict changes in groundwater levels. Because these algorithms are rarely used, it is both a difficulty and an opportunity for scholars and stakeholders in this domain to investigate new ML and MM techniques.

Both ML and MM techniques are anticipated to continue to deliver opportunities for modeling groundwater level changes. Researchers must begin adopting and adapting new approaches, as well as publishing widely, to ensure that research does not become static. Groundwater is important in people's daily lives, and it is threatened by global climate change. Researchers, governments, organizations, non-governmental organizations (NGOs), and other stakeholders are encouraged to advocate for climate change mitigation and education.

Our research has demonstrated that machine learning techniques are more recently used for modeling changes in groundwater levels than mathematical models. Random forest (RF), support vector machine (SVM), and artificial neural network (ANN) algorithms are the most popular machine learning (ML) approaches in literature. The most common predictive variables for changes in groundwater level are summarized based on our investigation (Table 4). Additionally, of the primary papers, 30.8 percent employ RMSE as

their primary evaluation metric, followed by MAE (10.6 percent), NSE (10.6 percent), and R (10.6 percent) (Table 6).

**Author Contributions:** Conceptualization and design, S.A. and T.Z.; methodology, S.A., T.Z., P.A. and V.V.; software, P.A.; validation, V.V., T.Z. and P.A.; formal analysis, S.A., T.Z., V.V. and P.A.; investigation, T.Z., V.V. and S.A.; resources, P.A., T.Z. and V.V.; data curation, S.A.; writing—original draft preparation, S.A., T.Z., P.A. and V.V.; writing—review and editing, T.Z., V.V. and P.A.; visualization, S.A.; supervision, T.Z., V.V. and P.A.; project administration, T.Z., P.A. and V.V. All authors have read and agreed to the published version of the manuscript.

**Funding:** This research received no external funding.

**Institutional Review Board Statement:** Not applicable.

**Informed Consent Statement:** Not applicable.

**Data Availability Statement:** Not applicable.

**Acknowledgments:** We wish to express our sincere gratitude to Adwoa Afriyie for her encouragements and advice given to us.

**Conflicts of Interest:** The authors declare no conflict of interest.

## Appendix A. List of Primary Studies (PS) in this Literature

Appendix A provides the list of primary studies (PS) used in this literature.

[PS1]   Q. Liu et al., "Simulation of regional groundwater levels in arid regions using interpretable machine learning models," *Sci. Total Environ.*, vol. 831, no. 2022, p. 154902, 2022.

[PS2]   H. Part, "Machine Learning Algorithms for soil". pp. 1–17, 2017.

[PS3]   L. Shi, H. Gong, B. Chen, and C. Zhou, "Land subsidence prediction induced by multiple factors using machine learning method," *Remote Sens.*, vol. 12, no. 24, pp. 1–17, 2020.

[PS4]   A. T. M. S. Rahman, T. Hosono, J. M. Quilty, J. Das, and A. Basak, "Multiscale groundwater level forecasting: Coupling new machine learning approaches with wavelet transforms," *Adv. Water Resour.*, vol. 141, no. May, 2020.

[PS5]   Z. Gaffoor, A. Gritzman, K. Pietersen, N. Jovanovic, A. Bagula, and T. Kanyerere, "An autoregressive machine learning approach to forecast high-resolution groundwater-level anomalies in the Ramotswa/North West/Gauteng dolomite aquifers of Southern Africa," *Hydrogeol. J.*, vol. 30, no. 2, pp. 575–600, 2022.

[PS6]   P. J. Jyolsna, B. V. N. P. Kambhammettu, and S. Gorugantula, "Application of random forest and multi-linear regression methods in downscaling GRACE derived groundwater storage changes," *Hydrol. Sci. J.*, vol. 66, no. 5, pp. 874–887, 2021.

[PS7]   Z. Kayhomayoon, F. Babaeian, S. G. Milan, and N. A. Azar, "A Combination of Metaheuristic Optimization Algorithms and Machine Learning Methods Improves the Prediction of Groundwater Level," pp. 1–25, 2022.

[PS8]   B. D. B. J. M. S. mohamed M. Morsey, "Forecasting Groundwater Table in a Flood Prone Coastal City with Long Short-term Memory and Recurrent Neural Networks," 2019.

[PS9]   A. P. Dadhich, R. Goyal, and P. N. Dadhich, "Assessment and Prediction of Groundwater using Geospatial and ANN Modeling," *Water Resour. Manag.*, vol. 35, no. 9, pp. 2879–2893, 2021.

[PS10]   E. Jeihouni, M. Mohammadi, S. Eslamian, and M. J. Zareian, "Potential impacts of climate change on groundwater level through hybrid soft-computing methods: a case study—Shabestar Plain, Iran," *Environ. Monit. Assess.*, vol. 191, no. 10, 2019.

[PS11]   M. E. Miro, D. Groves, B. Tincher, J. Syme, S. Tanverakul, and D. Catt, "Adaptive water management in the face of uncertainty: Integrating machine learning, groundwater modeling and robust decision making," *Clim. Risk Manag.*, vol. 34, no. May, p. 100383, 2021.

[PS12]   S. Ali et al., "Improving the resolution of grace data for spatio-temporal groundwater storage assessment," *Remote Sens.*, vol. 13, no. 17, 2021.

[PS13]   H. Li et al., "Spatiotemporal modeling of land subsidence using a geographically weighted deep learning method based on PS-InSAR," *Sci. Total Environ.*, vol. 799, p. 149244, 2021.

[PS14]   W. M. S. D. K. A. M. Milewski, "Downscaling GRACE TWSA data into high-resolution groundwater level anomaly using machine learning-based models in a glacial aquifer system," *MDPI*, 2019.

[PS15]   H. Han, B. Shi, and L. Zhang, "Prediction of landslide sharp increase displacement by SVM with considering hysteresis of groundwater change," *Eng. Geol.*, vol. 280, no. August 2020, p. 105876, 2021.

[PS16]   S. Sithara, P. S K, and S. G. Thampi, "Impact of projected climate change on seawater intrusion on a regional coastal aquifer," *J. Earth Syst. Sci.*, vol. 129, no. 1, 2020.

[PS17]   Z. Jiang et al., "Can ensemble machine learning be used to predict the groundwater level dynamics of farmland under future climate: a 10-year study on Huaibei Plain," *Environ. Sci. Pollut. Res.*, no. 0123456789, 2022.

[PS18]   A. Wunsch, T. Liesch, and S. Broda, "Deep learning shows declining groundwater levels in Germany until 2100 due to climate change," *Nat. Commun.*, vol. 13, no. 1, pp. 1–13, 2022.

[PS19]   D. Liu, A. K. Mishra, Z. Yu, H. Lü, and Y. Li, "Support vector machine and data assimilation framework for Groundwater Level Forecasting using GRACE satellite data," *J. Hydrol.*, vol. 603, no. PA, p. 126929, 2021.

[PS20]   S. Sahoo, T. A. Russo, J. Elliott, and I. Foster, "Machine learning algorithms for modeling groundwater level changes in agricultural regions of the U.S.," *Water Resources Research*, vol. 53, no. 5. pp. 3878–3895, 2017.

[PS21]   X. Yang, H. Zhang, and H. Zhou, "A Hybrid Methodology for Salinity Time Series Forecasting Based on Wavelet Transform and NARX Neural Networks," *Arab. J. Sci. Eng.*, vol. 39, no. 10, pp. 6895–6905, 2014.

[PS22]   S. Sachdeva and B. Kumar, "A novel ensemble model of automatic multilayer perceptron, random forest, and ZeroR for groundwater potential mapping," *Environ. Monit. Assess.*, vol. 193, no. 11, 2021.

[PS23]   K. Sun, L. Hu, J. Guo, Z. Yang, Y. Zhai, and S. Zhang, "Enhancing the understanding of hydrological responses induced by ecological water replenishment using improved machine learning models: A case study in Yongding River," *Sci. Total Environ.*, vol. 768, p. 145489, 2021.

[PS24]   A. Manandhar, H. Greeff, P. Thomson, R. Hope, and D. A. Clifton, "Shallow aquifer monitoring using handpump vibration data," *J. Hydrol. X*, vol. 8, no. May, 2020.

[PS25]   W. Yin, Z. Fan, N. Tangdamrongsub, L. Hu, and M. Zhang, "Comparison of physical and data-driven models to forecast groundwater level changes with the inclusion of GRACE—A case study over the state of Victoria, Australia," *J. Hydrol.*, vol. 602, no. July, 2021.

[PS26]   J. Chang, G. Wang, and T. Mao, "Simulation and prediction of supraper-mafrost groundwater level variation in response to climate change using a neural network model," *J. Hydrol.*, vol. 529, pp. 1211–1220, 2015.

[PS27]   E. A. O. O. S. S. T. Y. Reygadas, "Climate Change, Land Use/Land Cover Change, and Population Growth as Drivers of Groundwater Depletion in Central Valleys, Oaxaca, Mexico," 2019.

[PS28]   A. K. Sales, E. Gul, M. J. S. Safari, H. Ghodrat Gharehbagh, and B. Vaheddoost, "Urmia lake water depth modeling using extreme learning machine-improved grey wolf optimizer hybrid algorithm," *Theor. Appl. Climatol.*, vol. 146, no. 1–2, pp. 833–849, 2021.

[PS29]   M. S. A. K. A. D. A. Kar, "On predictability of Groundwater Level in Shallow Wells Using Satellite Observations," pp. 507–514, 2018.

[PS30]   N. Iqbal et al., "Groundwater Level Prediction Model Using Correlation and Difference Mechanisms Based on Boreholes Data for Sustainable Hydraulic Resource Management," *IEEE Access*, vol. 9, no. July, pp. 96092–96113, 2021.

[PS31]   A. M. M. M. B. T. W. M. S. T. C. Rasmussen, "Spatial Downscaling of GRACE TWSA Data to Identify Spatiotemporal Groundwater Level Trends in the Upper Floridan Aquifer, Georgia, USA," *2019*.

[PS32]   S. S. Malekzadeh M., Kardar S., "Simulation of groundwater level using MODFLOW, extreme learning machine and Wavelet-Extreme Learning Machine models," *Groundw. Sustain. Dev.*, 2019.

[PS33]   B. Yadav, P. K. Gupta, N. Patidar, and S. K. Himanshu, "Ensemble modelling framework for groundwater level prediction in urban areas of India," *Sci. Total Environ.*, vol. 712, p. 135539, 2020.

[PS34]   A. Amaranto, F. Munoz-Arriola, G. Corzo, D. P. Solomatine, and G. Meyer, "Semi-seasonal groundwater forecast using multiple data-driven models in an irrigated cropland," *J. Hydroinformatics*, vol. 20, no. 6, pp. 1227–1246, 2018.

[PS35]   F. Li, H. Gong, B. Chen, C. Zhou, and L. Guo, "Analysis of the contribution rate of the influencing factors to land subsidence in the eastern beijing plain, China based on extremely randomized trees (ERT) method," *Remote Sens.*, vol. 12, no. 18, pp. 1–21, 2020.

[PS36]   E. C. J. M. J. X. jia; S. Kraatz, "Identifying Subsurface Drainage using Satellite Big Data and Machine Learning via Google Earth Engine," 2019.

[PS37]   R. Khatibi and A. A. Nadiri, "Inclusive Multiple Models (IMM) for predicting groundwater levels and treating heterogeneity," *Geosci. Front.*, vol. 12, no. 2, pp. 713–724, 2021.

[PS38]   S. S. Band et al., "Comparative analysis of artificial intelligence models for accurate estimation of groundwater nitrate concentration," *Sensors (Switzerland)*, vol. 20, no. 20, pp. 1–23, 2020.

[PS39]   Omid Rahmati; Ali Golkarian; Trent Biggs, "Land subsidence hazard modeling: Machine learning to identify predictors and the role of human activities," *J. Environ. Manage.*, 2019.

[PS40]   Q. Cheng, F. Zhong, and P. Wang, "Baseflow dynamics and multivariate analysis using bivariate and multiple wavelet coherence in an alpine endorheic river basin (Northwest China)," *Sci. Total Environ.*, vol. 772, p. 145013, 2021.

[PS41]   R. Q. Gonzalez and J. J. Arsanjani, "Prediction of groundwater level variations in a changing climate: A danish case study," *ISPRS Int. J. Geo-Information*, vol. 10, no. 11, 2021.

[PS42]   A. Akter and S. Ahmed, "Modeling of groundwater level changes in an urban area," *Sustain. Water Resour. Manag.*, vol. 7, no. 1, 2021.

[PS43]   F. B. Banadkooki *et al.*, "Enhancement of Groundwater-Level Prediction Using an Integrated Machine Learning Model Optimized by Whale Algorithm," *Nat. Resour. Res.*, 2020.

[PS44]   S. Lee, K. K. Lee, and H. Yoon, "Using artificial neural network models for groundwater level forecasting and assessment of the relative impacts of influencing factors," *Hydrogeol. J.*, vol. 27, no. 2, pp. 567–579, 2019.

[PS45]   A. Khedri, N. Kalantari, and M. Vadiati, "Comparison study of artificial intelligence method for short term groundwater level prediction in the northeast Gachsaran unconfined aquifer," *Water Sci. Technol. Water Supply*, vol. 20, no. 3, pp. 909–921, 2020.

[PS46]   S. K. S. S. T. S. K. Roy, "Groundwater potentiality mapping using ensemble machine learning algorithms for sustainable groundwater management," 2021.

[PS47]   M. K. M. M. K. H. U. P. Jagtap, "USE OF MACHINE LEARNING IN GROUND-WATER LEVEL FORECASTING," *Int. J. Adv. Agric. Sci. Technol.*, vol. 7, no. 6, pp. 275–293, 2022.

[PS48]   S. R. C. D. P. louise Ryan, "Forecasting Multiple Groundwater Time Series with Local and Global Deep Learning Networks," *Int. J. Environ. Res. Public Health*, 2022.

[PS49]   M. Farzin, M. Avand, H. Ahmadzadeh, M. Zelenakova, and J. P. Tiefenbacher, "Assessment of ensemble models for groundwater potential modeling and prediction in a karst watershed," *Water (Switzerland)*, vol. 13, no. 18, 2021.

[PS50]   N. Djurovic *et al.,* "Comparison of Groundwater Level Models Based on Artificial Neural Networks and ANFIS," *Sci. World J.*, vol. 2015, 2015.

[PS51]   S. Bedi, A. Samal, C. Ray, and D. Snow, "Comparative evaluation of machine learning models for groundwater quality assessment," *Environ. Monit. Assess.*, vol. 192, no. 12, 2020.

[PS52]　E. Krogulec, J. J. Małecki, D. Porowska, and A. Wojdalska, "Assessment of causes and effects of groundwater level change in an urban area (Warsaw, poland)," *Water (Switzerland)*, vol. 12, no. 11. pp. 1–17, 2020.

[PS53]　E. A. O. Olivares *et al.*, "Climate change, land use/land cover change, and population growth as drivers of groundwater depletion in the Central Valleys, Oaxaca, Mexico," *Remote Sens.*, vol. 11, no. 11, pp. 1–25, 2019.

[PS54]　E. N. Sierikova and E. A. Strelnikova, "Mathematical Modeling of Groundwater Level Changing with Considering Evapotranspiration Factor," *Int. J. Mod. Stud. Mech. Eng.*, vol. 6, no. 1, pp. 19–25, 2020.

[PS55]　L. X. Li T., Song H., Huang G., Bi Y., "Assessment of groundwater changing trends through the generalized large well method with confined–unconfined flow model in open-pit mine area," 2014.

[PS56]　M. Szydłowski, W. Artichowicz, and P. Zima, "Analysis of the water level variation in the polish part of the vistula lagoon (Baltic sea) and estimation of water inflow and outflow transport through the strait of baltiysk in the years 2008–2017," *Water (Switzerland)*, vol. 13, no. 10, 2021.

[PS57]　A. M. F. T. C. T. S.-C. K. B. S. N. M. A. D. Rastogi, "Conjunctive management of surface and groundwater resources under projected future climate change scenerios," *J. Hydrol.*, 2016.

[PS58]　H. F. Abd-Elhamid, "Investigation and control of seawater intrusion in the Eastern Nile Delta aquifer considering climate change," *Water Sci. Technol. Water Supply*, vol. 17, no. 2, pp. 311–323, 2017.

[PS59]　Y. H.-D. Huang C.-S., Yang S.-Y., "Groundwater flow to a pumping well in a sloping fault zone unconfined aquifer," *2014*.

[PS60]　L. Li, D. A. Barry, C. B. Pattiaratchi, and G. Masselink, "Beachwin: Modelling groundwater effects on swash sediment transport and beach profile changes," *Environ. Model. Softw.*, vol. 17, no. 3, pp. 313–320, 2002.

[PS61]　M. A. Iyalomhe F., Rizzi J., Pasini S., Torresan S., Critto A., "Regional Risk Assessment for climate change impacts on coastal aquifers," 2015.

[PS62]　T. Strzelecki and M. Bartlewska-Urban, "Numerical calculations of the consolidation of flotation weste landfill 'Żelazny Most' based on Biot's model with the Kelvin-Voight rheological skeleton," *Arch. Civ. Eng.*, vol. 57, no. 2, pp. 199–213, 2011.

[PS63]　M. Person *et al.*, "Hydrologic response of the Crow Wing Watershed, Minnesota, to mid-Holocene climate change," *Bull. Geol. Soc. Am.*, vol. 119, no. 3–4, pp. 363–376, 2007.

[PS64]　C. He, T. Wang, Z. Zhao, Y. Hao, T. C. J. Yeh, and H. Zhan, "One-dimensional analytical solution for hydraulic head and numerical solution for solute transport through a horizontal fracture for submarine groundwater discharge," *J. Contam. Hydrol.*, vol. 206, no. January, pp. 1–9, 2017.

[PS65]　L. S. Andersen P.F., "A post audit of a model-designed ground water extraction system," 2003.

[PS66]　Y. C. Lin, C. S. Huang, and H. Der Yeh, "Analysis of Unconfined Flow Induced by Constant Rate Pumping Based on the Lagging Theory," *Water Resources Research*, vol. 55, no. 5. pp. 3925–3940, 2019.

[PS67]　M. J. M. Aguilera H., "The effect of possible climate change on natural groundwater recharge based on a simple model: A study of four karstic aquifers in SE Spain," 2009.

[PS68]　I. Quino-Lima *et al.*, "Spatial dependency of arsenic, antimony, boron and other trace elements in the shallow groundwater systems of the Lower Katari Basin, Bolivian Altiplano," *Sci. Total Environ.*, vol. 719, p. 137505, 2020.

[PS69]　B. I. Dvorak, M. Morley, and P. Denning, "Relative impact on GAC usage rates of operating strategies for treatment of contaminated groundwater," *Pract. Period. Hazardous, Toxic, Radioact. Waste Manag.*, vol. 12, no. 2, pp. 60–69, 2007.

[PS70]　S. R. J. Giambastiani B.M.S., Antonellini M., Oude Essink G.H.P., "Saltwater intrusion in the unconfined coastal aquifer of Ravenna (Italy): A numerical model," 2007.

[PS71]　A. Islam and H. A. Biswas, "Optimal Planning and Management of Groundwater Level Declination: A Mathematical Model," no. Lenhart 2007, 2019.

[PS72]　C. L. C. Isla F.I., Quiroz Londoño O.M., "Groundwater characteristics within loessic deposits: the coastal springs of Los Acantilados, Mar del Plata, Argentina". 2018.

[PS73]　K. H. Lee, N. Mizutani, D. S. Hur, and A. Kamiya, "The effect of groundwater on topographic changes in a gravel beach," *Ocean Eng.*, vol. 34, no. 3–4, pp. 605–615, 2007.

[PS74]　Y. O. Xun Z., Chuanxia R., Yanyan Y., Bin F., "Tidal effects of groundwater levels in the coastal aquifers near Beihai, China," 2006.

[PS75]　D. Pulido-Velazquez, A. Sahuquillo, J. Andreu, and M. Pulido-Velazquez, "A general methodology to simulate groundwater flow of unconfined aquifers with a reduced computational cost," *J. Hydrol.*, vol. 338, no. 1–2, pp. 42–56, 2007.

[PS76]　J. Almedeij and F. Al-Ruwaih, "Periodic behavior of groundwater level fluctuations in residential areas," *J. Hydrol.*, vol. 328, no. 3–4, pp. 677–684, 2006.

[PS77]　Z. Chen, S. E. Grasby, and K. G. Osadetz, "Relation between climate variability and groundwater levels in the upper carbonate aquifer, southern Manitoba, Canada," *J. Hydrol.*, vol. 290, no. 1–2, pp. 43–62, 2004.

[PS78]　R. Mahinroosta and L. Senevirathna, "The effectiveness of PFAS management options on groundwater quality in contaminated land using numerical modelling," *Chemosphere*, vol. 279, p. 130528, 2021.

[PS79]　van der P. M. Vissers M.J.M., "The stability of groundwater flow systems in unconfined sandy aquifers in the Netherlands," 2008.

[PS80]　L. Naji, M. Tawfiq, and A. K. Jabber, "Mathematical Modeling of Groundwater Flow," *C) Glob. J. Eng. Sci. Res.*, vol. 3, no. 10, pp. 2348–8034, 2016.

[PS81]　N. N. Dimitriou E., Moussoulis E., Stamati F., "Modelling hydrological characteristics of Mediterranean Temporary Ponds and potential impacts from climate change," 2009.

[PS82]　K. Kenda, J. Peternelj, N. Mellios, D. Kofinas, M. Čerin, and J. Rožanec, "Usage of statistical modeling techniques in surface and groundwater level prediction," *J. Water Supply Res. Technol. AQUA*, vol. 69, no. 3, pp. 248–265, 2020.

[PS83]　E. Serikova, E. Strelnikova, and V. Yakovlev, "Mathematical Model of Dangerous Changing the Groundwater Level in Mathematical Model of Dangerous Changing the Groundwater Level in Ukrainian Industrial Cities," no. January, 2015.

[PS84]　B. A.G., "Numerical modelling of salt-water intrusion due to human activities and sea-level change in the Godavari Delta, India," 2006.

[PS85]　C. Z. Y. W. X. C. B. Li, "Simulation of effects of groundwater level on vegetation change by combining FEFLOW software," 2004.

[PS86]　L. H. Chen T.-F., Wang X.-S., Wan L., "Analytical solutions of travel time to a pumping well with variable evapotranspiration," 2014.

[PS87]　Z. C. Wang X., Sun Y., Xu Z., Zheng J., "Feasibility prediction analysis of groundwater reservoir construction based on GMS and Monte Carlo analyses: a case study from the Dadougou Coal Mine, Shanxi Province, China," 2020.

[PS88]　Y. H.-D. Lin Y.-C., "A Lagging Model for Describing Drawdown Induced by a Constant-Rate Pumping in a Leaky Confined Aquifer," p. 11000, 2017.

[PS89]　H. M., "Accurate approximate semi-analytical solutions to the Boussinesq groundwater flow equation for recharging and discharging of horizontal unconfined aquifers," 2019.

[PS90]　V. R. M. Guevara Morel C.R., "Systematic investigation of non-Boussinesq effects in variable-density groundwater flow simulations," *J. Contam. Hydrol.*, 2015.

[PS91]　Y. X. Z. Z. mingwei L. C. Zhuang, "Analytical Solutions for Unsteady Groundwater Flow in an Unconfined Aquifer under Complex Boundary Conditions," 2019.

[PS92]　B. T.J., "Three-dimensional deformation and strain induced by municipal pumping, Part 2: Numerical," 2006.

[PS93]　X. Z. Q. D. V. C. X. Wang, "A hierarchical Bayesian model for decomposing the impacts of human activities and climate change on water resource in China". Science of the Total Environment, 2019.

[PS94]   H. G. B. M. Steinbuch L., Brus D.J., "Mapping the probability of ripened subsoils using Bayesian logistic regression with informative priors," 2018.

[PS95]   K. S. Gunawardhana L.N., "Statistical and numerical analyses of the influence of climate variability on aquifer water levels and groundwater temperatures: The impacts of climate change on aquifer thermal regimes," 2010.

[PS96]   M. R. S. Geng X., Boufadel M.C., Xia Y., Li H., Zhao L., Jackson N.L., "Numerical study of wave effects on groundwater flow and solute transport in a laboratory beach," 2014.

[PS97]   Y. F. Wang W., Zhao G., Li J., Hou L., Li Y., "Experimental and numerical study of coupled flow and heat transport," 2011.

[PS98]   A. A. S. A. J. Adeloye, "Mathematical modelling of effects of Irawan irrigation project water abstarctions on Murzuq aquifer systems in Libya," *J. Arid Environ.*, 2007.

[PS99]   T. Ondovčin, J. Mls, and L. Herrmann, "Mathematical Modeling of Tidal Effects in Groundwater," *Transp. Porous Media*, vol. 95, no. 2, pp. 483–495, 2012.

[PS100]   M. Malekzadeh, S. Kardar, and S. Shabanlou, "Simulation of groundwater level using MODFLOW, extreme learning machine and Wavelet-Extreme Learning Machine models," *Groundw. Sustain. Dev.*, vol. 9, p. 100279, 2019.

[PS101]   H. K. Moghaddam, H. K. Moghaddam, Z. R. Kivi, M. Bahreinimotlagh, and M. J. Alizadeh, "Developing comparative mathematic models, BN and ANN for forecasting of groundwater levels," *Groundw. Sustain. Dev.*, vol. 9, no. January, p. 100237, 2019.

[PS102]   J. B. Mohapatra, P. Jha, M. K. Jha, and S. Biswal, "Efficacy of machine learning techniques in predicting groundwater fluctuations in agro-ecological zones of India," *Sci. Total Environ.*, vol. 785, p. 147319, 2021.

[PS103]   R. Barzegar, E. Fijani, A. Asghari Moghaddam, and E. Tziritis, "Forecasting of groundwater level fluctuations using ensemble hybrid multi-wavelet neural network-based models," *Sci. Total Environ.*, vol. 599–600, pp. 20–31, 2017.

[PS104]   J. Shiri, O. Kisi, H. Yoon, K. K. Lee, and A. Hossein Nazemi, "Predicting groundwater level fluctuations with meteorological effect implications-A comparative study among soft computing techniques," *Comput. Geosci.*, vol. 56, pp. 32–44, 2018.

[PS105]   X. Huang, L. Gao, R. S. Crosbie, N. Zhang, G. Fu, and R. Doble, "Groundwater recharge prediction using linear regression, multi-layer perception network, and deep learning," *Water (Switzerland)*, vol. 11, no. 9, 2019.

[PS106]   J. Scibek, D. M. Allen, A. J. Cannon, and P. H. Whitfield, "Groundwater-surface water interaction under scenarios of climate change using a high-resolution transient groundwater model," *J. Hydrol.*, vol. 333, no. 2–4, pp. 165–181, 2007.

[PS107]   C. Guevara-Ochoa, A. Medina-Sierra, and L. Vives, "Spatio-temporal effect of climate change on water balance and interactions between groundwater and surface water in plains," *Sci. Total Environ.*, vol. 722, p. 137886, 2020.

[PS108]   M. Gedeon, I. Wemaere, and J. Marivoet, "Regional groundwater model of north-east Belgium," *J. Hydrol.*, vol. 335, no. 1–2, pp. 133–139, 2007.

[PS109]   A. D. M. Scibek J., "Modeled impacts of predicted climate change on recharge and groundwater levels," 2000.

[PS110]   L. Dong, J. Chen, C. Fu, and H. Jiang, "Analysis of groundwater-level fluctuation in a coastal confined aquifer induced by sea-level variation," *Hydrogeol. J.*, vol. 20, no. 4, pp. 719–726, 2012.

[PS111]   J. Dams, S. T. Woldeamlak, and O. Batelaan, "Predicting land-use change and its impact on the groundwater system of the Kleine Nete catchment, Belgium," *Hydrol. Earth Syst. Sci.*, vol. 12, no. 6, pp. 1369–1385, 2008.

[PS112]   Y. Al-Zu' bi, M. Shatanawi, O. Al-Jayoussi, and A. Al-Kharabsheh, "Application of Decision Support System for Sustainable Management of Water Resources in the Azraq Basin—Jordan," *Water Int.*, vol. 27, no. 4, pp. 532–541, 2002.

[PS113]   M. Jiang, S. Xie, and S. Wang, "Water use conflict and coordination between agricultural and wetlands—a case study of Yanqi basin," *Water (Switzerland)*, vol. 12, no. 11, pp. 1–18, 2020.

[PS114]   Y. Liu *et al.*, "Development and Application of a Water and Salt Balance Model for Well-Canal Conjunctive Irrigation in Semiarid Areas with Shallow Water Tables," *Agric.*, vol. 12, no. 3, 2022.

[PS115]   A. Huo, X. Wang, Y. Liang, C. Jiang, and X. Zheng, "Integrated numerical model for irrigated area water resources management," *J. Water Clim. Chang.*, vol. 11, no. 4, pp. 980–991, 2020.

[PS116]   A. K. Chaudhry, M. A. Alam, and K. Kumar, "Groundwater contamination monitoring and modeling for a part of Satluj river basin," *Desalin. Water Treat.*, vol. 212, pp. 152–163, 2021.

[PS117]   J. Zhang et al., "Impact of Mountain Reservoir Construction on Groundwater Level in Downstream Loess Areas in Guanzhong Basin, China," *Water (Switzerland)*, vol. 14, no. 9, pp. 1–12, 2022.

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
