# Peer review of "Mathematical and Machine Learning Models for Groundwater Level Changes: A Systematic Review and Bibliographic Analysis"

_futureinternet, doi:10.3390/fi14090259_

Round 1

Reviewer 1 Report

The manuscript entitled “Mathematical and machine learning models for ground water level changes: a systematic review and bibliographic analysis” analyzed the previous studies in the field of groundwater level changes. The manuscript does not have a suitable structure and therefore cannot help readers in order to choose the appropriate method and to understand the weakness and strengths of different models. I have several concerns that should be considered by authors to improve the manuscript:

- It is necessary to explain the novelty of the present research clearly in the last paragraph of the introduction section.

- Machine learning approach is a wide field of knowledge, demanding to explain in detail for the journal readers. Also, the differences of various applied methods in groundwater level estimation should be analyzed.

- The authors mentioned ensembles and hybrid models. Since these methods have been widely-used in the latest studies due to their high accuracy, it is suggested describing further these methods in the text.

- There is no suitable link between different parts of this manuscript which may confuse the readers. The structure of the manuscript should be modified.

- Please improve the conclusion section by writing a brief description of the whole manuscript, with more focus on the special findings of this manuscript.

- There are some grammatical problems in the text. The whole manuscript should be checked.

Author Response

We (the authors) express our profound gratitude for taking your time to review our paper. We appreciate your effort and suggestions made to enrich our study. We have also address the issues you raised and believe to meet your kindest consideration.

Sincerely yours. 

Reviewer 2 Report

The paper entitled MATHEMATICAL AND MACHINE LEARNING MODELS FOR GROUNDWATER LEVEL CHANGES: A SYSTEMATIC REVIEW AND BIBLIOGRAPHIC ANALYSIS is connected with literature review of the different methods for hydrogeological research. It is a very important issue. The paper is based on 117 articles, what suggests a good review. In the introduction, please elaborate on why the determination of groundwater level changes is so important.

Author Response

(The authors gave the same response as above.)

Reviewer 3 Report

A lot of work, a lot of statistics, but we don't know who will use this effort, apart from the statistics hunters and the competitors regarding the most sophisticated models, the most sophisticated calculation algorithms etc.
We believe, each researcher knows, best, the specifics of the studied area, the necessary statistics and will apply the most appropriate modeling or simulation methods from the literature, or will develop their own methodologies to obtain the most accurate result, according to the purpose and with local, zonal or regional requirements.
It is, at least, unelegant to recommend something to someone, even from the top of a ranking, without knowing the specifics of their desires and problems, which they are trying to solve.
If the models or algorithms developed do not bring any benefit or do not solve a problem of the research funding society, apart from the pride of being very sophisticated, appreciated and well classified, then there is a question mark. Maybe those financial resources could be used more efficiently.
In addition, it is difficult to understand, where is that component so scientific, bringing great news in the literature useful to someone, who solves a problem or brings a benefit to someone other than the authors. We see, only, a statistic, a thematic and bibliographical statistic, achievable even by scientists of less performance, but very diligent and readers of scientific literature.
Beyond the above, the use of phrases such as those related to rows 120-124, 142-145, 160-161, 181-184, 217-227, may be inappropriate.
Perhaps it would be better to let researchers find the most appropriate methods to fill methodological and modeling gaps. Only, the great personalities of world science are suitable to make recommendations for the use of methodologies for specific fields or requirements and, even they, do not know all the specificities and local laws so diverse in the world ...

Author Response

(The authors gave the same response as above.)

Reviewer 4 Report

The paper is mostly devoted to combined presentation of topical review on modelling of the state of groundwater and information retrieval tricks which were useful for authors during processing of a large number of various research papers.  It is interesting but there is a major concern which cannot lead to its acceptance.

However, it is really unclear to me how such a paper may fit the topic of the special issue `` High Performance Computing for AI Applications ''. In my opinion this paper is not devoted to HPC at all even though it considers a really interesting field of AI appications (but does not offer any novel algorithmic solutions). Hence, I think it should not be accepted for this special issue and requires to be shifted for consideration at regular issue of the journal or even other topical journal (e.g. MDPI Water).

I should also add that groundwater modelling is interesting not only from the side of its level but also state of the water (e.g. concentrations of different chemicals and geospatial analysis) is also very important, see e.g.

[1] Shadrin D. et al. An Automated Approach to Groundwater Quality Monitoring—Geospatial Mapping Based on Combined Application of Gaussian Process Regression and Bayesian Information Criterion //Water. – 2021

[2] Razavi-Termeh, S. Vahid, Abolghasem Sadeghi-Niaraki, and Soo-Mi Choi. "Groundwater potential mapping using an integrated ensemble of three bivariate statistical models with random forest and logistic model tree models." Water 11.8 (2019): 1596.

and there are numerous references and even dedicated journals...

Author Response

(The authors gave the same response as above.)

Round 2

Reviewer 1 Report

The authors reply my comments and I recommend publication of this manuscript.

Reviewer 4 Report

The authors have responded to my questions corretly, as well as editors have taken into account my concern about the topical issue for this paper. Hence, after these actions I think that the paper may be accepted but maybe requires additional proofreading.

I am hoping that this topical review will be useful for many researchers in future.